



# Four North American glaciers advanced past their modern positions thousands of years apart in the Holocene

Andrew G. Jones[1], Shaun A. Marcott[1], Andrew L. Gorin[2], Tori M. Kennedy[3], Jeremy D. Shakun[2], Brent M. Goehring[3], Brian Menounos[4,5], Douglas H. Clark[6], Matias Romero[1], Marc W. Caffee[7,8]

[1]Department of Geoscience, University of Wisconsin-Madison, Madison, Wisconsin, USA
[2]Department of Earth & Environmental Sciences, Boston College, Chestnut Hill, Massachusetts, USA
[3]Department of Earth and Environmental Sciences, Tulane University, New Orleans, Louisiana, USA
[4]Geography Department, University of Northern British Columbia, Prince George, British Columbia, Canada
[5]Hakai Institute, Campbell River, British Columbia, Canada
[6]Department of Geology, Western Washington University, Bellingham, Washington, USA
[7]Department of Physics and Astronomy, Purdue University, West Lafayette, IN 47907, USA
[8]Department of Earth, Atmospheric, and Planetary Science, Purdue University, West Lafayette, IN 47907, USA

*Correspondence to:* Andrew G. Jones (agjones3@wisc.edu)

**Abstract.** The global retreat of alpine glaciers provides visible evidence of industrial-era warming, but how glacier position today compares to glacier length fluctuations over the Holocene is less clear. Glaciers in North America advanced over the Holocene, occupying their maximum Holocene position in the late 19th century before rapidly retreating to their sizes today. We assess when four North American glaciers, located between 38–60° N, were larger or smaller than their modern (2018–2020 CE) sizes during the Holocene. We measure 26 paired cosmogenic *in situ* [14]C and [10]Be concentrations in recently exposed proglacial bedrock and applied a Monte Carlo forward model to reconstruct plausible bedrock exposure-burial histories. We find that these glaciers advanced past their modern sizes thousands of years apart during the Holocene: a glacier in the Juneau Ice Field (BC, Canada) by ~2 ka, Kokanee Glacier (BC, Canada) at ~6 ka, and Mammoth Glacier (WY, USA) at ~1 ka; the fourth glacier, Conness Glacier (CA, USA), was larger than its modern size for the duration of the Holocene until present. The disparate Holocene exposure-burial histories are at odds with expectations of similar glacier histories given the presumed shared climate forcings of decreasing Northern Hemisphere summer insolation through the Holocene followed by global greenhouse gas forcing in the industrial era. We interpret the range in histories to be the result of unequal amounts of modern retreat relative to each glacier's Holocene length history, rather than asynchronous Holocene advance histories. The intensity and rate of modern warming has exacerbated length differences between glaciers that occur due to hypsometry and response time. We hypothesize that highly varying magnitudes of glacier change in North America today is a departure from similar magnitudes of glacier change over the Holocene.

## 1 Introduction

As global temperatures have increased since the industrial era, alpine glaciers have receded worldwide to lengths not observed since record-keeping began in the 17th century (Oerlemans, 2005). Declining glacial meltwater poses threats to agriculture, human consumption, and biodiversity (Milner et al., 2017; Huss et al., 2017) with severe consequences for people who live in high mountain regions (IPCC, 2021). Although significant strides have been made towards projecting glacier disappearance (e.g. Rounce et al., 2023), there is ample room to reduce uncertainty in these projections by using past fluctuations to better understand glacier change. Temperatures in the 21st century are almost as warm (Marcott et al., 2013) or warmer than (Osman et



al., 2021) the peak temperatures of the Holocene. How glaciers fluctuated in the Holocene provides critical insight into glacier
change in the coming decades as glaciers equilibrate to modern climate.

        In North America, expanded ice positions are well-constrained, but there are few measurements of glaciers in recessed
positions (Solomina et al., 2015) in which to compare to today. Glaciers in the western United States retreated from valley
positions during deglaciation to largely within their cirques by the start of the Holocene (Marcott et al., 2019); this retreat is

recorded in a series of moraines dated to be deglacial and earliest Holocene in age. Then, over the Holocene, glaciers in western
Canada advanced from minimum positions occupied between 11–7 ka (Menounos et al., 2009), when Northern Hemisphere
summer insolation was high, to maximum positions in the latest Holocene, during the so-called Little Ice Age (LIA, 1350–1850
CE; Wanner et al., 2008; Menounos et al., 2009). While Holocene maximum positions are recorded in well-preserved LIA
moraines, how small glaciers became during the early Holocene is difficult to assess—as the glaciers advanced from their

minimum positions, they overrode their prior moraines, destroying the record of their former size (Gibbons et al., 1984).
Inferences of glacier activity are commonly made from glaciogenic sediment fluxes recorded in distal lakes, but these do not
directly record glacier length. Lake records have been paired with radiocarbon dating of fossil trees in moraines to constrain
periods of glacier advance, with such a multi-proxy approach forming much of the understanding of Holocene glacier trends that
exists today (Davis et al., 2009; Solomina et al., 2015; Menounos et al., 2009; Osborn et al., 2007). In many mid-to-low latitude

locations, fossil trees are absent in moraines, and lake records alone are used to infer relative changes. The overall weaker
constraints on glacier position in the early to mid-Holocene have prevented a direct assessment of modern glacier size within a
Holocene context.

        An additional consideration in studying glaciers is how their length incorporates climate. We expect that glaciers across
western North America advanced roughly synchronously over the Holocene because the first-order climate controls are similar:

insolation decreases across the northern mid-latitudes and mean annual temperature decreases coherently across the western
United States (Osman et al., 2021; Menounos et al., 2009). Glacier length is also a robust record of climate, having among the
highest known signal-to-noise ratios in recording industrial era changes (Roe et al., 2017). Yet, other variables can cause
variation in glacier length responses, such as glacier hypsometry and climate heterogeneity (Hugonnet et al., 2021; Rupper et al.,
2009). For example, in the western United States, glaciers in the Sierra Nevada, California are thought to have disappeared from

~10–3 ka prior to 'Neoglacial' cooling (Bowerman and Clark, 2011; Cary, 2018; Konrad and Clark, 1998; Porter and Denton,
1967), while glaciers in the Teton Range, Wyoming, appear resilient, persisting through the warmth of the early to mid-Holocene
as rock glaciers (Larsen et al., 2020). Substantial debris coverage in the Tetons may have mitigated ablation during peak early
Holocene summer warmth, as put forth by Larsen et al. (2020), allowing glaciers to persist. Another possibility is that the same
change in equilibrium line altitude (ELA) across the western United States caused glacier disappearance in one range and not in

another as a function of the accumulation area lost (Porter, 2000). In either case, interpreting glacier length records across
regions without direct measurements of past size has proven difficult.

        Here, we use the $^{14}$C-$^{10}$Be chronometer in proglacial bedrock (Goehring et al., 2011; Vickers et al., 2020) to directly
compare modern glacier positions to glacier length fluctuations over the Holocene at four North American glaciers. Cosmogenic
nuclides accumulated in bedrock collected from the terminus of modern ice reflect past intervals when the glacier was larger or

smaller than its present-day position. The bedrock samples 'outline' the terminus of the glacier today and provide a direct spatial
comparison between past and present glacier length, addressing the lack of spatial data prior to the LIA. Our approach attempts
to minimize the challenges of comparing glaciers with unique hypsometry across many degrees of latitude by reconstructing
Holocene glacier length fluctuations relative to the common reference point of glacier size today. We go to four glaciers from




38–60° N to survey change across western North America. Using *in situ* $^{14}$C-$^{10}$Be cosmogenic exposure dating of proglacial
bedrock and a Monte Carlo-based forward model (Vickers et al., 2020), we assess not only integrated exposure and burial
durations of recently exposed bedrock, but *when* the glaciers were larger or smaller than today. Our findings indicate more
heterogeneity amongst the four sites than expected given their relative proximity to one another and shared climate forcings. We
suggest that rather than representing disparate Holocene histories, non-uniform amounts of industrial era retreat caused some
glaciers to recede more than others relative to their LIA maxima. We propose that the large differences in the magnitude of
glacier retreat across western North America is a departure from relatively synchronous advance and retreat histories over the
Holocene.

**2 Glacier Setting and Historical Retreat**
The four North American glaciers in this study are located along the American Cordillera (Figure 1). We mapped the LIA glacier
extents from moraine and trimlines, an intermediate position using the Randolph Glacier Inventory (RGI Consortium, 2017), and
the modern (2021 CE) position for each glacier. These glaciers retreated 14–70% from their maximum Holocene extents inferred
from the nearest end moraine and trimlines (Figure 2). Glaciers in the western United States and Canada began retreating from
their LIA moraines by 1880 CE (Basagic and Fountain, 2011; DeVisser and Fountain, 2015; Menounos et al., 2009). The amount
of retreat since 1880 is related to the climate change at the glacier and the glacier's response time, defined as the time for each
glacier's length to reach equilibrium with climate change. Hypsometric variables such as ice surface slope, cirque-wall shading,
and debris cover impact response time, with steep ice surface slopes thought to be a particular correlate for quick response times
(Pelto and Hedlund, 2001; Zekollari et al., 2020). Below, we summarize each glacier's modern retreat history, hypsometry, and
Holocene paleoclimate information, with data reported in Table 1. We approximate each glacier's response time as maximum
mass loss from the terminus divided by mean glacier thickness (Jóhannesson et al., 1989) as shown in Eq. 1:

$$\tau = \frac{H}{-b_t} \tag{1}$$

where τ = time for volume adjustment in years, H = thickness in meters (m), and $b_t$ = maximum mass loss at the terminus in m
yr$^{-1}$. This approximation for response time (along with glacier slope) provides insight into how much each glacier has been
impacted by industrial-era warming. We caveat this response time as being a minimum estimate of how quickly a glacier could
adjust its volume because we use the maximum mass loss from the terminus; its purpose is to serve as a common point of
comparison.
From north to south, the first glacier is an unnamed valley glacier in the Juneau Ice Field (henceforth JIF Glacier) in the
Coast Mountains on the border of southeast Alaska and British Columbia. It is the largest glacier studied here, with an area of
15.2 km$^2$ in 2021 (compared to the smallest glacier presented, Conness Glacier, at 0.1 km$^2$). It has the lowest-sloping ice surface
of the four glaciers (5°) and has the slowest response time (τ = 27). It is the most impacted by absolute area loss from climate
change, but the least impacted by percentage relative to its Holocene maximum at 70% of its LIA area. The 2004 area of JIF
glacier mapped in Figure 1 is likely similar to the area of the glacier at ~2 ka prior to the latest Holocene and LIA advances
(Clague et al., 2010). JIF Glacier's Holocene history is expected to exhibit the 'classic' two-phase history observed in Alaska and
western Canada of smaller in the early Holocene and larger in the late Holocene (Davis et al., 2009; McKay and Kaufman, 2009;
Levy et al., 2004; Menounos et al., 2009).



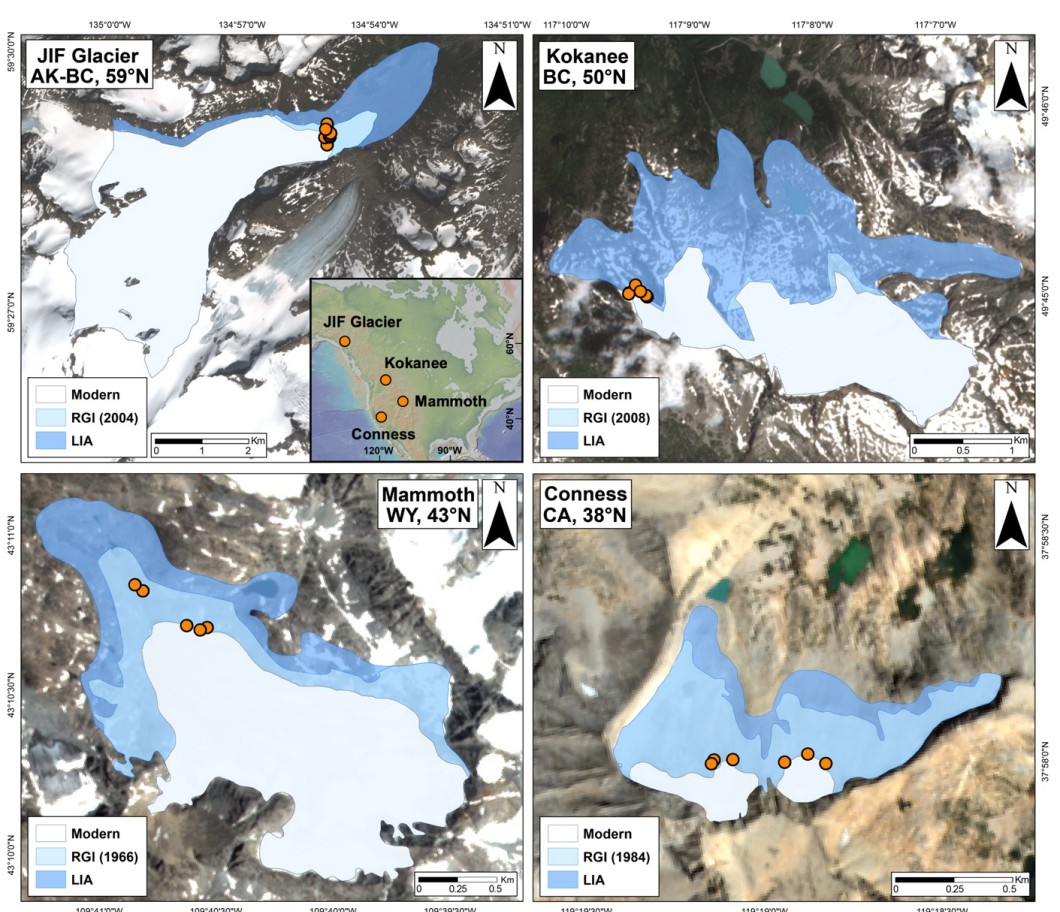

**Figure 1.** Study site locations and historical ice extent. Little Ice Age extent mapped from trimlines and moraines assumed to be last occupied by glaciers in ~1880 CE (Menounos et al., 2009; Wanner et al., 2008). Proglacial bedrock sample locations denoted by orange dots. Modern ice extent represents glacier size in 2021 CE. Randolph Glacier Inventory (RGI) date depends upon data availability in the RGI database. Juneau Ice Field Glacier (JIF) Glacier: Coast Mountains, Alaska-British Columbia border. Kokanee Glacier: Selkirk Mountains, British Columbia. Mammoth Glacier: Wind River Range, Wyoming. Conness Glacier: Sierra Nevada, California. Image credit: Copernicus.





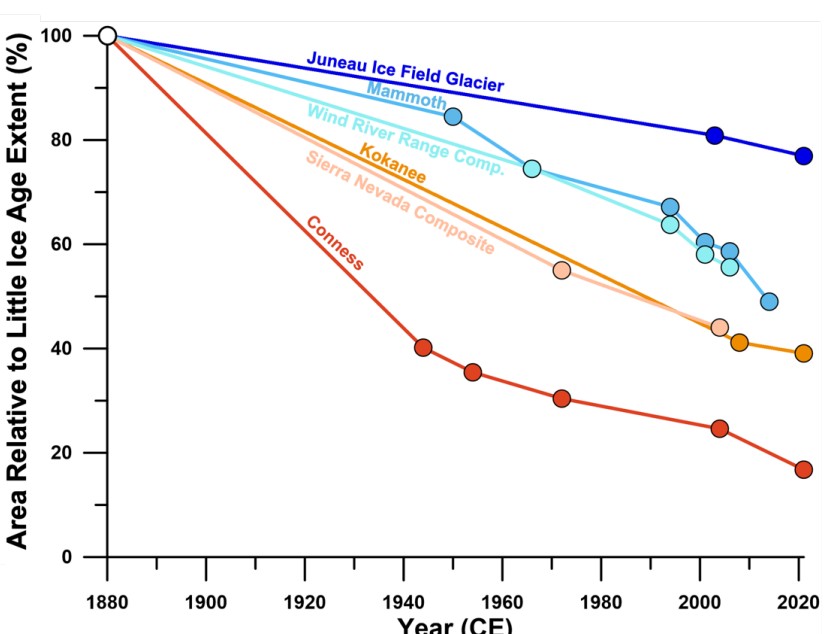

**Figure 2.** Modern glacier area (2021) relative to Little Ice Age extent. Little Ice Age extents assumed to be last occupied at ~1880 CE (Menounos et al., 2009; Wanner et al., 2008). Two composites of regional glaciers from the Wind River Range and Sierra Nevada are included to provide context for regional glacier change (DeVisser and Fountain, 2015; Basagic and Fountain, 2011). Internal mapping has been supplemented with mapping by DeVisser and Fountain (2015) at Mammoth Glacier and Basagic and Fountain (2011) at Conness Glacier. See Supplement for further details on area calculation.

**Table 1. Glacier hypsometry data**

| Glacier | Latitude (DD) | Longitude (DD) | Modern Area (km²) | Little Ice Age Area[a] (km²) | % of LIA Area | Response Time[b] (yrs) | Modern Ice Surface Slope (°) | Aspect (°) | Mean Elev. (m) |
|---------|---------------|----------------|-------------------|------------------------------|---------------|------------------------|------------------------------|------------|----------------|
| JIF | 59.474 | -134.964 | 10.7 | 15.2 | 70 | 27 | 5 | 53 | 1490 |
| Kokanee | 49.749 | -117.142 | 2.6 | 5.5 | 46 | 14 | 22 | 22 | 2560 |
| Mammoth | 43.169 | -109.667 | 1.7 | 3.4 | 49 | 23 | 9 | 318 | 3630 |
| Conness | 37.968 | -119.318 | 0.1 | 0.6 | 14 | - | 23 | 11 | 3600 |

[a]Little Ice Age areas were mapped for this study based on trimline and moraines as seen in Figure 1.
[b]Calculated according to Jóhannesson et al., 1989. Mean glacier thickness from Farinotti et al., 2019. Maximum mass loss at the terminus from Hugonnet et al., 2021. Conness Glacier omitted due to lack of suitable data. See Table S6 in the Supplement for calculation details.

Kokanee Glacier is in the Selkirk Mountains of southeastern British Columbia and occupies 46% of its LIA area. Its ice-surface slope is 22°, the second-highest slope, and it has the second quickest response time ($\tau = 14$). Glaciers in the Selkirk Mountains are similarly expected to match evidence in western Canada of a minimum position between 11–7 ka followed by numerous advances from 7 ka until the LIA (Menounos et al., 2009).



Mammoth Glacier is in the Wind River Range, Wyoming and is 49% of its LIA maximum area, which is similar to the average of glaciers from the Wind River Range (Figure 2; DeVisser and Fountain, 2015). It has the second-lowest ice-surface slope at 9° and the second-slowest response time ($\tau = 23$). Cores from distal lakes in the valley below Mammoth Glacier suggest that the glacier has been active since at least 4.5 ka—though perhaps disappearing in the early Holocene—but was much smaller

than its LIA extent from 4.5–1 ka (Davies, 2011), implying a substantial advance during the LIA.

Conness Glacier is in the Sierra Nevada, California, USA. It is the smallest glacier in this study with an area of 0.1 km$^2$ in 2021, representing 14% of its LIA maximum extent (0.6 km$^2$). Conness has the steepest ice surface slope, 23°, suggesting the quickest response time of the glaciers studied here. Response time calculations are omitted for Conness Glacier because the RGI mapping is too different from glacier size today (over 2x the area) to provide an accurate estimate. Its LIA area loss of 70% is the

highest of the four glaciers presented, and is higher than the average for Sierra Nevada glaciers (~40%, Basagic and Fountain, 2011, Figure 2). Conness Glacier is thought to have disappeared between 10–3 ka before reforming and advancing to its LIA extent based on distal lake glaciogenic sediment flux records (Konrad and Clark, 1998).

### 3 Materials and Methods


#### 3.1 Field sampling, laboratory procedures, and cosmogenic nuclide concentration measurements

We collected surface exposure samples by hammer and chisel from glacially abraded, quartz-bearing bedrock within tens of meters of the modern ice margin. We targeted bedrock high points (local topographic maxima) as they are less likely to have prior till accumulation during episodes of exposure. We sampled abraded bedrock to minimize the change that the bedrock had

been quarried, while acknowledging that abrasion does not exclude the possibility of subglacial quarrying prior to abrasion (Rand and Goehring, 2019).

Quartz separation procedures are modified from Kohl and Nishiizumi (1992). We crushed and sieved samples (~1 kg of rock) to the 250–710 µm size fraction, performed magnetic separation, and etched them in dilute HCl and HF/HNO$_3$. We performed chemical froth flotation on non-magnetic grains to remove feldspathic minerals. We etched remaining grains in dilute

(1-5%) HF/HNO$_3$ until pure quartz was reached. Samples were wet-sieved at 250 µm to remove lingering mafic minerals, fine quartz grains, and partially dissolved feldspars. Quartz purity was confirmed by ICP-OES measurement. Quartz was separated at Boston College for samples from Kokanee Glacier and Conness Glacier, at the University of Wisconsin-Madison for samples from Mammoth Glacier, and at Tulane University for samples from JIF Glacier.

We analyzed all samples for *in situ* cosmogenic $^{14}$C and $^{10}$Be concentrations. $^{10}$Be extraction procedures follow

Ceperley et al. (2019). We extracted $^{10}$Be at the University of Wisconsin-Madison for all samples except JIF Glacier samples, which were extracted at Tulane University and follow procedures of Ditchburn and Whitehead (1994). We spiked the samples with a $^9$Be carrier prepared from raw beryl (OSU White standard, $251.6 \pm 0.9$ ppm; JIF Glacier samples TuBE, $904 \pm 28$ ppm). We isolated Be from Fe, Ti, Al, and other ions using anion-cation exchange chromatography. We precipitated BeOH in a pH 8 solution and incinerated the BeOH gels at ~1000°C to convert to BeO. We packed samples with Nb powder into stainless steel

cathodes for accelerator mass spectrometry analysis. $^{10}$Be/$^9$Be ratios were measured at Purdue Rare Isotope Measurement Lab relative to 07KNSTD dilution series. We ran each batch with at least two blanks; we background-correct samples using the average blank value from within the specific sample batch.

We extracted $^{14}$C at Tulane University for all samples following Goehring et al. (2019). $^{14}$C/$^{13}$C ratios were measured at the National Ocean Sciences Accelerator Mass Spectrometry facility. We calculated exposure ages using the CRONUS Earth-





online calculator v.3 using LSDn scaling (Balco et al., 2008; Lifton et al., 2014). Exposure age calculations have ± 4% external

uncertainty (Phillips et al., 2016).

### 3.2 $^{14}$C-$^{10}$Be ratios as records of past glacier length

We determine the total time during the Holocene that each glacier was larger or smaller than its present-day size using *in situ*

cosmogenic $^{14}$C-$^{10}$Be dating of proglacial bedrock (Goehring et al., 2011). The bedrock abutting the terminus of a glacier today is

exposed or buried during the Holocene as the glacier advances and retreats over the bedrock (Figure 3). We collected several (n

= 6–9) bedrock samples that were recently exposed during modern retreat (see supplement for field photos and historical

imagery). We assume that all samples experienced the same exposure-burial history because the samples have been exposed by

retreating ice contemporaneously. Exposure duration is inferred from $^{10}$Be concentrations which accumulate when the glacier is

smaller than its modern size; burial duration is inferred from disparity between $^{14}$C and $^{10}$Be concentrations because $^{14}$C

($t_{1/2}$=5,700 ±30 yr) decays rapidly during burial relative to $^{10}$Be ($t_{1/2}$=1.387 ±0.012 Myr) (Chmeleff et al., 2010; Hippe, 2017).

We assume that subglacial erosion during the last glacial period removed any nuclides from pre-Holocene exposure (Ivy-Ochs

and Briner, 2014), consistent with the absence of cosmogenic exposure ages greater than the length of the Holocene measured in

this study. We also assume that burial by till or nonglacial debris (e.g. rockfalls, hillslope sloughing) is insignificant. Although it

is possible that bedrock sites were covered in debris during intervals of past ice recession, that scenario is unlikely to uniformly

affect all samples given the spatially variability of such deposits, and thus coherency between samples is a reasonable check.

Results are plotted as nuclide concentrations along surface exposure and burial isochrons (Figure 4), but we note these are

merely a visual aid for interpreting the nuclide ratios; more complex modeling is needed to fully understand the relationship

between exposure, burial, and erosion, which we assess with the Monte Carlo forward model. Our assumption of simultaneous

exposure (or burial) implies that differences in measured concentrations are due to erosion rather than glacier histories at each

sample.

### 3.3 Monte Carlo forward model of nuclide concentrations in proglacial bedrock

To constrain *when* exposure and burial occurred, beyond integrated durations, we apply a Monte Carlo forward model of $^{14}$C and

$^{10}$Be concentrations in proglacial bedrock (Vickers et al., 2020). The purpose is to iterate through millions of combinations of

nuclide production, radioactive decay, and glacial erosion to identify Holocene exposure-burial scenarios that best explain the

measured nuclide concentrations in our proglacial bedrock samples. Fundamentally, the model simulates how $^{14}$C and $^{10}$Be

concentrations evolve as a glacier advances and retreats over a transect of surface bedrock samples, simulating production during

exposure, and decay and erosion when buried.

205         The Monte Carlo forward model applies 100,000 unique exposure-burial histories to theoretical bedrock columns. Each

exposure-burial scenario is a sequence of 100-year timesteps that represent either exposure or burial. There are 110 total

timesteps summing to 11,000 years, representing the Holocene when glaciers reached roughly modern sizes (Marcott et al.,

2019). The scenarios include a range of natural variability intended to simulate natural glacier length oscillations, where some

stochastic length variation over bedrock is expected. At each timestep, the bedrock is either 1) exposed at the surface, with

nuclides produced along a 5 m bedrock depth profile, or 2) buried by ice, where nuclide production ceases and the uppermost

bedrock is eroded. Ice thickness during burial is assumed to be greater than 10 m where production becomes negligible

(Goehring et al., 2011). Erosion rates from 0–2.5 mm yr$^{-1}$ are applied to each scenario in steps of 0.1 mm yr$^{-1}$, such that each of

the 100,000 scenarios is tested 26 times. Radioactive decay occurs throughout the simulation, depressing $^{14}$C-$^{10}$Be ratios relative



to the production ratio during burial. Scenarios that recreate measured concentrations in a bedrock sample within 2-σ uncertainty

are considered plausible. Overlapping scenarios—scenarios that work for *all* sample sites at a given glacier—are recorded and

saved as a viable exposure-burial history. The result is a list of exposure-burial scenarios (at varying erosion rates) that can

recreate the observed nuclide concentrations for all samples at a given glacier. Solutions that do not include burial in the final

200 years are discarded given the broad evidence of expanded glacier positions during the LIA and satellite imagery showing

most sample positions buried under ice until the last few years before sampling (Figures S1–S4). The overlapping scenarios are

then averaged together to generate the probability of exposure at each timestep. Samples with $^{14}$C-$^{10}$Be ratios above the

production ratio for surface exposure are excluded from the Monte Carlo forward model because they are theoretically

impossible, though the possibility of high erosion rates exhuming nuclides produced in the subsurface is explored in the

discussion.

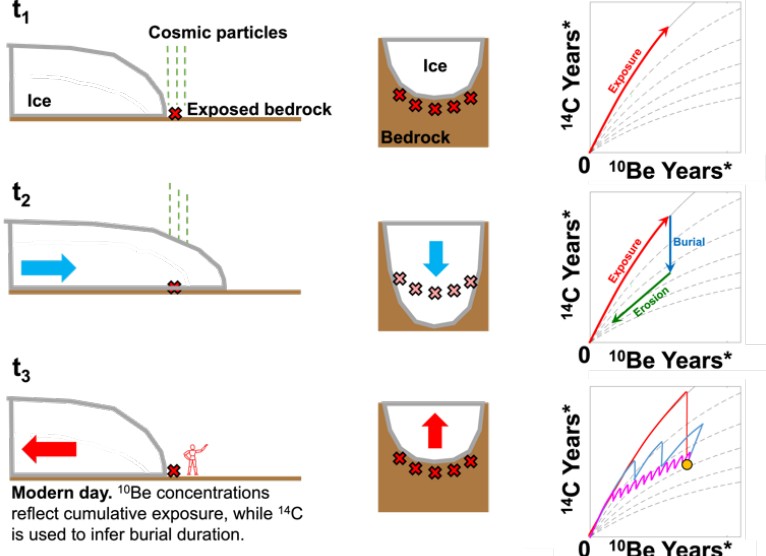

**Figure 3.** Schematic of *in situ* $^{14}$C-$^{10}$Be concentrations in proglacial bedrock in response to glacier length fluctuations. At $t_1$,

bedrock is exposed and nuclides accumulate along the continuous exposure curve. At $t_2$, the glacier has advanced and buried the

bedrock. Nuclide production ceases while erosion occurs; $^{14}$C decays rapidly relative to $^{10}$Be during burial, while erosion

removes both nuclides. At $t_3$, bedrock is re-exposed during modern retreat and sampled. Many scenarios of exposure, burial, and

erosion can explain the measured nuclide concentrations, represented by the colored lines in the nuclide plot. We apply a Monte

Carlo forward model to identify best-fitting scenarios.

## 4 Results

$^{14}$C-$^{10}$Be sample ratios at JIF Glacier, Kokanee Glacier, and Mammoth Glacier require a complex history of exposure and burial,

while nuclide concentrations from Conness Glacier are very low, near AMS-system blank values (Figure 4 and Table 2).

Samples from each glacier have distinct $^{14}$C-$^{10}$Be ratios that cluster in different regions of the isochron plot, suggesting unique



Holocene exposure-burial histories at each glacier rather than a uniform history amongst all four sites. CRONUS exposure ages are mapped onto satellite imagery of the glacier forefields (Figure 5). $^{10}$Be exposure ages, which provide insight into total Holocene exposure of the bedrock, are highest at Mammoth Glacier, with a mean age of $7.9 \pm 1.5$ ka (standard deviation) from the three highest-concentration samples. Glacial erosion removes nuclides, so the highest nuclide concentrations best

approximate total Holocene exposure. Kokanee Glacier samples have the next highest $^{10}$Be exposure ages, with an average of $4.7 \pm 0.8$ ka amongst the three highest concentrations. JIF Glacier bedrock has a mean age $3.4 \pm 0.4$ ka of its three highest concentration samples. Conness Glacier bedrock has a mean age of $0.1 \pm 0.02$ ka for its three highest measurements.

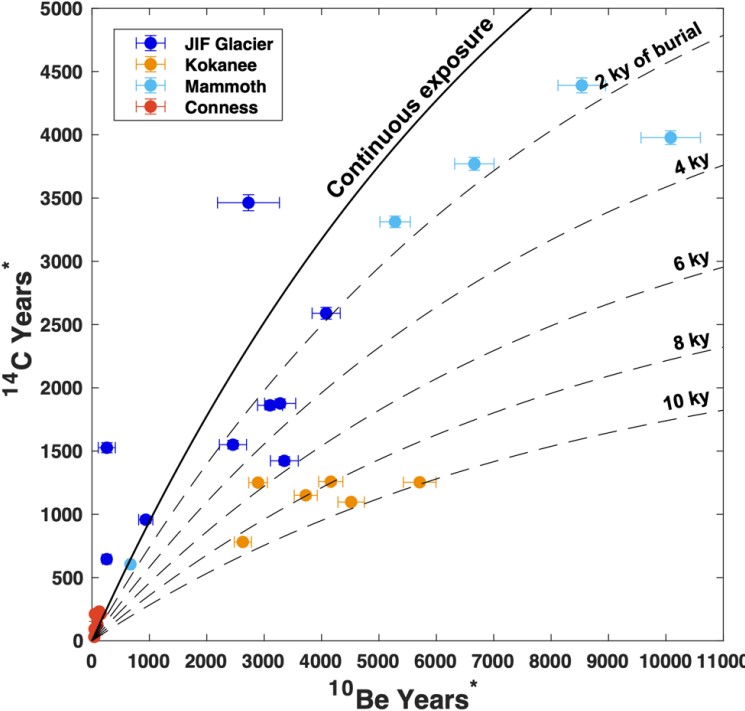

**Figure 4.** $^{14}$C-$^{10}$Be ratios plotted against burial isochrons. Nuclide concentrations have been normalized by their local production
rates to easily compare between sites (such that concentrations are expressed in years but are not true exposure ages). The solid line represents the evolution of $^{14}$C-$^{10}$Be ratios if samples are continuously exposed. Subsequent dashed lines represent $^{14}$C-$^{10}$Be ratios when sample bedrock is buried and the $^{14}$C-$^{10}$Be ratio is depressed through radioactive decay. The x-axis approximates integrated exposure during the Holocene in years, while the y-axis values inform burial duration. Erosion and radioactive decay are not incorporated into the nuclide concentrations; for concentrations modeled as exposure ages, see the CRONUS exposure
ages in Table 2.





**Table 2. $^{14}$C and $^{10}$Be concentrations and exposure ages. All uncertainties are 1σ.**

| Sample | $^{10}$Be[a] ($10^3$ atoms g$^{-1}$) | $^{10}$Be Uncert. ($10^3$ atoms g$^{-1}$) | $^{14}$C[a] ($10^3$ atoms g$^{-1}$) | $^{14}$C Uncert. ($10^3$ atoms g$^{-1}$) | $^{14}$C/$^{10}$Be | $^{10}$Be Age[b] (yr) | Uncert. (yr) | $^{14}$C Age[b] (yr) | Uncert. (yr) |
|---|---|---|---|---|---|---|---|---|---|
| *Juneau Ice Field (JIF) Glacier* | | | | | | | | | |
| JIF-01 | 39.60 | 3.27 | 75.3 | 1.42 | 1.90 | 3163 | 261 | 2154 | 46 |
| JIF-02 | 32.90 | 6.51 | 139.0 | 2.54 | 4.22 | 2708 | 536 | 4946 | 124 |
| JIF-03 | 3.14 | 1.80 | 61.6 | 1.60 | 19.62 | 250 | 143 | 1694 | 49 |
| JIF-04 | 48.30 | 2.90 | 102.0 | 1.83 | 2.11 | 3939 | 237 | 3206 | 70 |
| JIF-05 | 36.80 | 2.58 | 73.5 | 1.43 | 2.00 | 3090 | 217 | 2211 | 49 |
| JIF-06 | 3.01 | 8.74 | 25.3 | 1.02 | 8.42 | 249 | 72 | 662 | 28 |
| JIF-07 | 11.00 | 1.45 | 37.4 | 1.07 | 3.40 | 847 | 112 | 993 | 30 |
| JIF-08 | 29.30 | 2.82 | 61.5 | 1.32 | 2.10 | 2324 | 224 | 1732 | 41 |
| JIF-09 | 40.10 | 2.90 | 56.6 | 1.28 | 1.41 | 3294 | 238 | 1599 | 40 |
| *Kokanee Glacier* | | | | | | | | | |
| KG-01 | 81.12 | 4.52 | 109.0 | 1.87 | 1.35 | 2860 | 160 | 1385 | 26 |
| KG-02 | 74.43 | 4.22 | 68.9 | 1.34 | 0.93 | 2572 | 146 | 780 | 16 |
| KG-03 | 105.67 | 5.70 | 102.0 | 2.18 | 0.96 | 3707 | 200 | 1260 | 29 |
| KG-04 | 136.84 | 6.96 | 103.0 | 1.78 | 0.75 | 4523 | 230 | 1190 | 22 |
| KG-05 | 172.02 | 8.56 | 117.0 | 2.04 | 0.68 | 5562 | 277 | 1392 | 26 |
| KG-06 | 122.44 | 6.21 | 115.0 | 2.00 | 0.94 | 4133 | 210 | 1394 | 26 |
| *Mammoth Glacier* | | | | | | | | | |
| MG-01 | 334.55 | 17.26 | 568.0 | 7.71 | 1.70 | 6388 | 330 | 5566 | 108 |
| MG-02 | 504.01 | 25.85 | 597.0 | 8.10 | 1.18 | 9307 | 478 | 5910 | 117 |
| MG-03 | 422.73 | 20.54 | 653.0 | 8.83 | 1.55 | 7884 | 384 | 6610 | 137 |
| MG-04 | 31.95 | 2.68 | 86.8 | 1.78 | 2.72 | 574 | 48 | 578 | 12 |
| MG-05 | 252.02 | 12.59 | 476.0 | 6.50 | 1.89 | 5309 | 265 | 4826 | 89 |
| *Conness Glacier* | | | | | | | | | |
| CG-01 | 6.37 | 0.79 | 26.5 | 1.09 | 4.16 | 124 | 15 | 184 | 8 |
| CG-02 | 5.26 | 0.74 | 29.1 | 1.49 | 5.53 | 103 | 15 | 203 | 11 |
| CG-03 | 3.85 | 0.71 | 15.9 | 10.10 | 4.13 | 77 | 14 | 112 | 72 |
| CG-04 | 1.59 | 0.50 | 3.7 | 7.09 | 2.36 | 33 | 10 | 28 | 52 |
| CG-05 | 1.77 | 0.48 | 11.3 | 7.26 | 6.36 | 37 | 10 | 83 | 53 |
| CG-06 | 2.02 | 0.45 | 25.5 | 1.37 | 12.62 | 42 | 9 | 184 | 10 |

[a]Nuclide concentrations are blank-corrected specific to each sample batch. See supplement for background measurements. All AMS measurements standardized to 07KNSTD.
[b]Calculated using the CRONUS-Earth online calculator v.3 (Balco et al., 2008) with the LSDn scaling scheme (Lifton et al., 2014) and global production rate. Ages assume continuous exposure with no erosion, modern elevation, and standard atmosphere. Uncertainties are analytical (i.e., internal) only. Rock density is 2.65 g cm$^{-3}$. JIF Glacier samples collected in 2019; Kokanee Glacier in 2018; Mammoth Glacier in 2020; Conness Glacier in 2018.

The $^{14}$C-$^{10}$Be ratios plotted against burial isochrons in Figure 5 provide approximate bedrock exposure and burial durations at each glacier. Proceeding north to south, JIF Glacier samples have two populations: samples below the continuous exposure curve that mostly plot along a ~3 ky burial isochron (thus suggesting 3 ky of cumulative Holocene burial) and samples above the continuous exposure curve. Deep glacial erosion (0.5–3.0 m) via subglacial quarrying is a known way to elevate the $^{14}$C-$^{10}$Be ratio above the continuous exposure curve due to the depth-dependent production of $^{14}$C versus $^{10}$Be (Rand and Goehring, 2019). Bedrock samples from Kokanee Glacier have the lowest $^{14}$C-$^{10}$Be ratios of any of the four sites and suggest Holocene burial durations of 6–10 ky. Low ratios are only possible through decay of $^{14}$C relative to $^{10}$Be; this process dominates



when bedrock is shielded from production during burial. The Mammoth Glacier samples cluster around 1–2 ky of burial. These samples have high $^{14}$C-$^{10}$Be ratios which are only possible with minimal burial. Samples closer to the origin along the same isochron experienced more erosion than those with higher concentrations. The Conness Glacier samples have uniquely low nuclide concentrations amongst the four glaciers and thus plot near the origin. Samples with nuclide concentrations near the detection limit in both nuclides are only observed at Conness Glacier of the four glaciers presented here. Scatter in bedrock samples at each site is expected due to variable erosion depths along the glacier width.

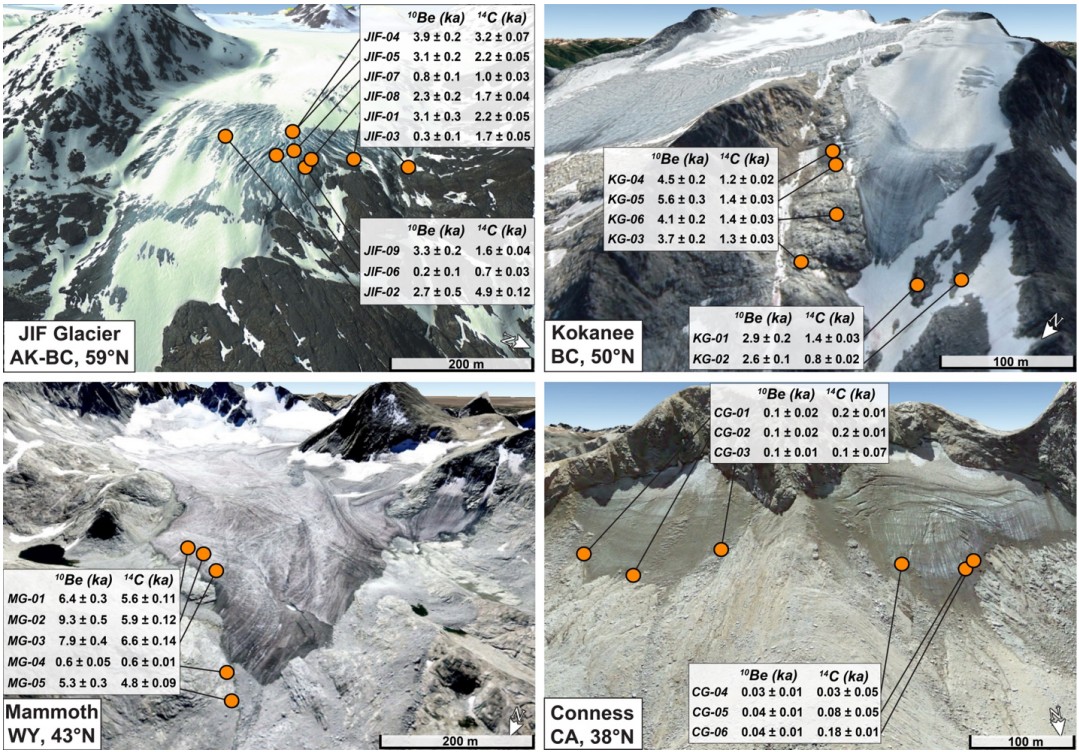

**Figure 5.** Oblique satellite images of glacier forefields with sample locations and $^{14}$C and $^{10}$Be exposure ages. Sample locations denoted by colored dots corresponding to each glacier. Images are from different years; the highest-resolution years with least snow cover were selected. Some samples are still buried by glacial ice at the time the image was captured, showing the recent burial prior to sampling in 2018-2020 CE. Image years: Juneau Ice Field Glacier, 2011. Kokanee Glacier, 2021 Mammoth Glacier, 2006. Conness Glacier, 2013. Image credit: © Google Earth.



The Monte Carlo forward model results predict that JIF, Kokanee, and Mammoth glaciers were smaller than their
modern size in the early-to-mid-Holocene and larger than their modern size in the late Holocene (Figure 6). The probability of bedrock exposure at each site exhibits a quick transition from exposure to burial. For simplicity, we interpret burial to begin at the mid-point of this transition and report the time in which glaciers advanced past modern positions as approximate values. JIF Glacier bedrock samples suggest advance past modern size at ~2 ka. However, the JIF Glacier bedrock lacks a probability of exposure greater than 70% from 8–3 ka, when exposure is most likely to have occurred. Kokanee Glacier advances past its
modern size at ~6 ka. These samples have tight agreement amongst overlapping scenarios: the probability of exposure is >90% in the early Holocene and drops to <10% by 5 ka. Mammoth Glacier advances past modern size shortly after ~1 ka. Mammoth Glacier samples have a high probability (80-90%) of exposure from 10–1 ka.

Conness Glacier samples are unique amongst the four glaciers in that nuclide concentrations cannot be successfully reproduced with our Monte Carlo analysis using any scenarios that include exposure unless an exceptionally high erosion rate
(>0.5 mm yr$^{-1}$) is applied (Figure S8). Erosion rates at bedrock samples from the other three glaciers are dominantly below 0.5 mm yr$^{-1}$, with all three glaciers expected to be more erosive than Conness, therefore we favor these results (Figure 6), but consider the possibility of high erosion below. Full model outputs are provided in the Supplement.

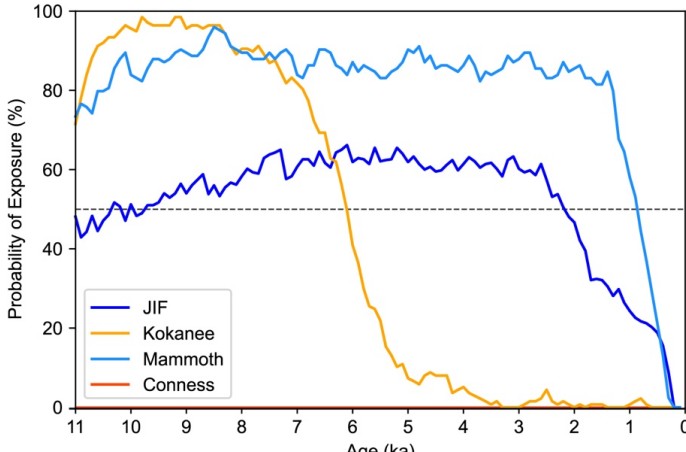

**Figure 6.** Modeled probability of exposure of bedrock at the four glaciers. The probability of exposure is the percentage of plausible exposure-burial scenarios that feature exposure at a given timestep. For example, if 900 out of 1000 plausible exposure-
burial scenarios include exposure from 8.1–8.0 ka, then the probability of exposure from 8.1–8.0 ka is 90%. We interpret the change in probability of exposure from greater than 50% to less than 50% to indicate roughly when the glacier advanced past its modern size and bedrock samples were buried. Note that Conness Glacier's probability of exposure is 0% based on our modeling, and thus its history is along the x-axis.

**5 Discussion**


In agreement with broad evidence of Holocene glacier advance in western North America (Menounos et al., 2009; Solomina et al., 2015; Davis et al., 2009), our results at all four glaciers are best explained by glacier expansion from the early to late




Holocene. The trend is apparent from the nuclide ratios alone: sample populations from JIF Glacier, Kokanee Glacier, and Mammoth Glacier all exhibit the depressed [14]C-[10]Be ratios characteristic of early-to-mid-Holocene exposure followed by late

Holocene burial. At Conness Glacier, ice growth cannot be directly inferred due to near-blank nuclide concentrations, but given evidence for increasing glaciogenic sediment flux in the late Holocene recorded in distal lakes directly downstream of Conness Glacier (Konrad and Clark, 1998) and elsewhere in the Sierras (Bowerman and Clark, 2011), it is highly likely Conness advanced in the late Holocene as well. Mid-to-late Holocene advances also agrees with similar studies using *in situ* [14]C-[10]Be to constrain glacier advance in the Swiss Alps, Peru, and Uganda (Goehring et al., 2011; Schimmelpfennig et al., 2022; Vickers et

al., 2020). Glacier expansion from the early to mid-Holocene supports the presence of a Holocene Thermal Maximum in western North America that remains persistent in proxy temperature records (Kaufman and Broadman, 2023), at odds with recent data-assimilation research that find slight warming in the Holocene (Osman et al., 2021). Sea-surface temperature reconstructions from 30–90° N (Marcott et al., 2013) and terrestrial temperature reconstructions from western North America (Routson et al., 2021) resemble decreasing Northern Hemisphere summer insolation (Figure 7). Increasing winter precipitation also likely

contributes to glacier growth. Whether the observed glacier expansion is due to glaciers' seasonal bias is unclear, but our results support the proxy data trends of cooler summers and wetter winters.

Where we find our results surprising, however, is in the non-uniform bedrock burial durations. The [14]C-[10]Be sample ratios and modeled exposure-burial histories suggest the glaciers advanced past their modern sizes thousands of years apart. Consider the two endmembers: Mammoth Glacier was likely smaller than its modern size for almost the entire Holocene until

the last millennium, while Conness Glacier was larger than its modern size for the entire Holocene until present. These are vastly different Holocene length histories relative to their terminus positions today, despite evidence that western North America experienced roughly similar climate changes over the Holocene (Shuman and Marsicek, 2016; Menounos et al., 2009). Although it is possible to interpret the range of bedrock burial durations as bellwethers of previously unrecognized climate heterogeneity over the Holocene, we instead consider non-uniform amounts of modern retreat relative to Holocene fluctuations. Our sampling

approach treats the modern position of each glacier as a common reference point, but modern glaciers are transiently responding to geologically abrupt warming; they are out of equilibrium with climate (IPCC, 2021). We posit that the non-uniform burial durations are the result of some glaciers having receded further relative to their long-term Holocene sizes than others. Below, we first evaluate the modeled probabilities of exposure and compare them with the [14]C-[10]Be ratios and extant paleoclimate data. Then, we explore hypotheses for why the modern glacier terminus positions are non-uniform relative to their Holocene length

histories and their implications.



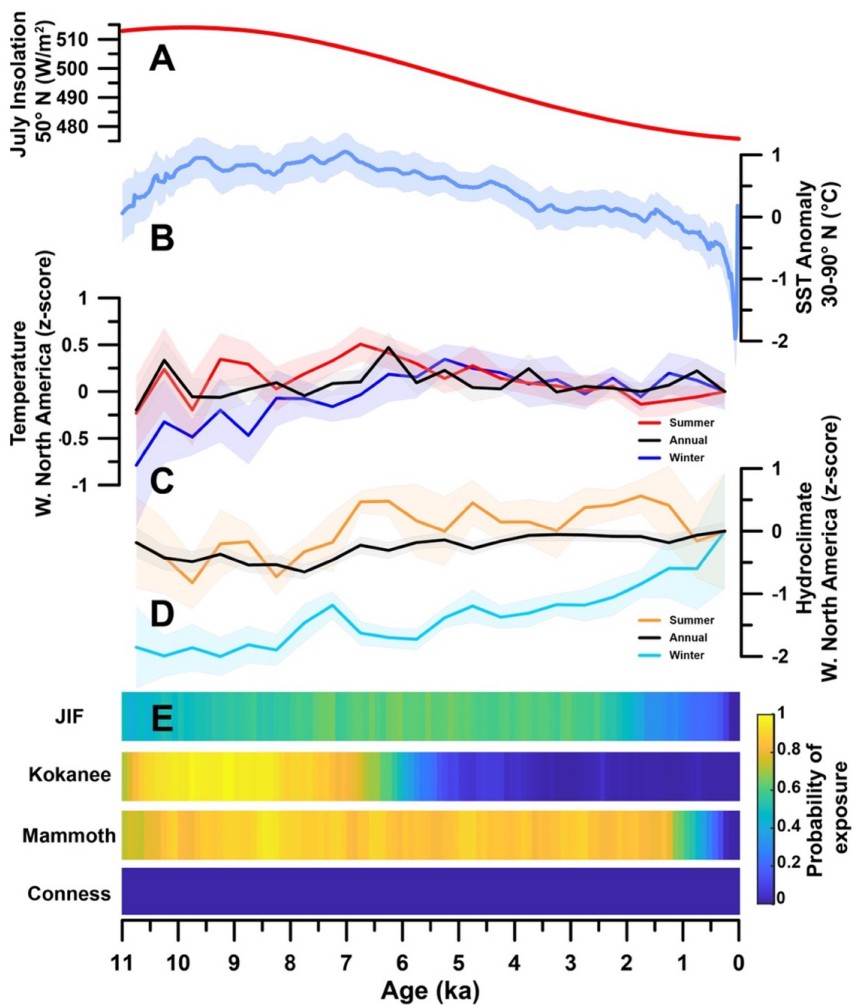

**Figure 7.** Holocene paleoclimate data and modeled probability of exposure at each glacier. **A.** 50° N July insolation (Laskar et al., 2004). **B.** Sea-surface temperature proxy reconstruction for 30–90° N where shaded band is 1-σ uncertainty (Marcott et al., 2013). **C, D.** Annual and seasonal temperature and hydroclimate reconstruction from western North America proxy data (Routson et al., 2021). **E.** Modeled probability of exposure for each glacier. Color bar ranges from 100% chance of exposure in yellow to 0% chance of exposure (i.e. burial) in blue.

### 5.1 Holocene exposure-burial histories at each glacier

At JIF Glacier, the measured concentrations and burial isochrons necessitate exposure in the early-to-mid-Holocene and burial in the late Holocene (Figure 4), but the overlapping scenarios from the Monte Carlo forward model do not fully agree on precisely when the exposure occurred. The highest $^{10}$Be exposure age is 3.9 ± 0.2 ky (sample JIF-04) which we interpret as the minimum Holocene exposure duration (minimum because erosion removes nuclides from a 'true' exposure duration). The population of



samples plotting along the ~3 ky burial isochron approximates burial duration. Summing the ~4 ky of exposure and ~3 ky of
burial, however, the total time exposed or buried falls short of the length of the Holocene. It is the exposure duration that appears
particularly anomalous: there is broad evidence from fossil trees in moraines that glaciers in the Juneau Ice Field did not advance
past their modern positions until the late Holocene (Clague et al., 2010; Menounos et al., 2009). One hypothesis to explain the
lower-than-expected exposure durations is that JIF Glacier deglaciated slowly, not reaching its minimum Holocene size until ~7
ka, akin to other arctic glaciers (Briner et al., 2016), instead of at ~10 ka like the mid-latitude glaciers in the western United
States and Canada. In this scenario, bedrock would be exposed from ~7–3 ka and buried from 3-0 ka, matching the simplified
exposure-burial durations from the isochron plot. However, these values do not consider erosion, and this interpretation would
contradict the inferred history of many other glaciers in the Coast Range that record minimum glacier extents between 11–7 ka
(Menounos et al., 2009).

       Instead, a preferred explanation for lower-than-expected exposure durations is that JIF Glacier eroded enough bedrock
to significantly alter $^{14}$C-$^{10}$Be ratios. $^{14}$C-$^{10}$Be ratios increase with depth as a function of $^{14}$C's higher production rate at depth
than $^{10}$Be. Near the bedrock surface (uppermost 10 cm of bedrock), the difference is negligible as $^{14}$C and $^{10}$Be are produced
similarly (Hippe, 2017). At greater bedrock depths, in the range of 50–300 cm, the $^{14}$C-$^{10}$Be ratio can become so high that it
exceeds the surface-production ratio. The Monte Carlo forward model is designed to test high-erosion scenarios and account for
changes in the $^{14}$C-$^{10}$Be ratio with depth. However, high erosion rates present challenges by simply increasing the number of
plausible scenarios: $^{14}$C-$^{10}$Be ratios near the continuous exposure ratio can be created by continuous exposure alone *or* by
exposure, burial, and high erosion. Low $^{14}$C-$^{10}$Be ratios are only possible through decay of $^{14}$C during burial, which numerically
constrains solutions.

       The lack of agreement amongst overlapping scenarios at JIF Glacier is almost certainly related to the multitude of ways
to recreate measured concentrations. Alaskan glaciers are among the most erosive in the world (Koppes and Montgomery, 2009).
Erosion rates of 1–15 mm yr$^{-1}$ have been observed in coastal Alaska over the last few centuries (Hallet, 2011), orders of
magnitude higher than our erosion rate estimates at Kokanee and Mammoth Glaciers (<0.5 mm yr$^{-1}$, excluding likely quarried
samples). The high erosion rates increase the number of plausible scenarios, and no singular plausible history presents itself.
High erosion rates also explain the apparent 'youngness' of $^{10}$Be ages, where primary bedrock erosion removed nuclides that
otherwise would show exposure from at least ~10–4 ka as predicted by paleoclimate evidence in western Canada. We suggest
JIF Glacier is uniquely erosive amongst the four glaciers studied here, more so than even the Monte Carlo forward model
predicts, and this causes the modeled exposure durations to be underestimated. We therefore interpret that JIF Glacier was larger
than its present-day size by ~2 ka as a minimum estimate.

       Kokanee Glacier samples converge tightly upon early Holocene exposure followed by burial from ~6 ka until modern
retreat (Figure 5). The samples have the lowest $^{14}$C-$^{10}$Be ratios of any population measured (Table 2), which agrees with the
longest burial duration inferring by forward modeling. The exceedingly low $^{14}$C-$^{10}$Be ratios mathematically limit possible
exposure-burial scenarios because $^{14}$C can only be decreased relative to $^{10}$Be via radioactive decay, which dominates during
burial. The strong agreement amongst the overlapping exposure-burial scenarios reflects the necessity of extensive burial to
produce the measured ratios. The advance of Kokanee Glacier across its modern size threshold at ~6 ka is earlier than any other
site studied with this method, but in good agreement with lake and moraine records that document glacier advances as early as 7
ka (Menounos et al., 2009). The successive glacier advances are inferred to be interspersed with episodes of glacier retreat. Our
results imply that the retreat events did not extend beyond the modern position of the glacier, suggesting that these retreats were
minimal in size if they occurred at all.



The high exposure ages of the Mammoth Glacier samples require the glacier to be smaller than its modern size for most of the Holocene. The mean [10]Be age of the three most exposed samples (7.9 ± 1.5 ky) implies extensive exposure, while the similarly high [14]C concentrations in these samples reflect low burial durations. Again, exposure-burial histories are mathematically constrained: the measured [14]C-[10]Be ratios are only possible with exposure for most of the Holocene. Monte Carlo forward model results support this interpretation, predicting Mammoth Glacier was smaller than its modern size from 10–1 ka before crossing its modern threshold in the last millennium. There are two outliers amongst the sample population: sample MG-04 has anomalously low exposure ages ([10]Be: 0.5 ± 0.05 ka, [14]C: 0.6 ± 0.01 ka) relative to the other samples, while sample MG-02 ([10]Be: 9.3 ± 0.5 ka; [14]C: 5.9 ± 0.1 ka) suggests more burial than the other sites (Figure 4). We test the possibility that either of these samples exerts an outsize influence on our results with a sensitivity test where we exclude MG-04 or MG-02 to see if results change (Figure S7). In results from both tests, exposure-burial histories still suggest extensive Holocene exposure. We interpret the low nuclide concentrations of MG-04 to be due to deep erosion, likely by subglacial quarrying, and the low [14]C-[10]Be ratio of MG-02 to be an artifact of the simplifications made for the isochron plot. Of the four glaciers in this study, Mammoth Glacier recorded the latest advance across its modern margin.

Interestingly, Mammoth Glacier today is ~50% of its LIA size, implying that in the last millennium, it nearly doubled in size. This interpretation agrees well with prior findings from distal lake sediment cores that Mammoth Glacier was much smaller than its LIA extent from 4.5–1 ka and significantly advanced after 1 ka (Davies, 2011). The non-linear behavior of Mammoth Glacier's length suggests ice dynamics may play a role in its length changes in addition to a more favorable climate for growth during the LIA. One hypothesis is that for most of the Holocene, the glacier was buttressed by a bedrock knob or contained within its cirque, and when a thickness threshold was crossed, the glacier spilled out of its cirque rapidly, manifesting as a doubling in size over such a short interval. It is hard to explain such a large change with climate alone, though it is theoretically possible.

The extremely low nuclide concentrations at Conness Glacier could be explained by 1) glacial burial in the Holocene until modern retreat, 2) deep erosion, or 3) burial by debris when bedrock would otherwise be exposed during glacier retreat. Holocene exposure followed by deep erosion is a plausible mechanism to explain the low nuclide concentrations. To test whether the low-nuclide concentrations represent burial or deep erosion, we ran two forward model simulations for Conness Glacier, one with erosion rates tested from 0–2.5 mm yr[-1] (like at the other glaciers) and one with erosion rates capped at 0.5 mm yr[-1] (Figure S8). When erosion rates up to 2.5 mm yr[-1] are included, solutions are found that feature mid to late-Holocene exposure followed by deep erosion. Erosion rates used were skewed toward higher values, with most plausible scenarios requiring erosion rates approaching 2.5 mm yr[-1]. When erosion is capped at 0.5 mm yr[-1], no solutions can recreate the measured concentrations at all samples.

What are reasonable erosion rates for alpine glaciers? Estimates in the literature of subglacial abrasion rates vary widely, from 0.1 mm yr[-1] (Rand and Goehring, 2019) up to 5 mm yr[-1] (Wirsig et al., 2016). However, modeled erosion rates at Kokanee and Mammoth Glacier are almost all below 0.5 mm yr[-1] (Figure S6). Similarly, 10 out of 11 proglacial bedrock samples at the Rhone Glacier in Switzerland—a comparable mid-latitude glacier—are less than 0.4 mm yr[-1] (Goehring et al., 2011). We therefore find it unlikely that erosion has removed sufficient nuclides exposure in the early Holocene. Subglacial quarrying is a plausible mechanism for deep erosion, but quarrying is spatially heterogeneous (Woodard et al., 2019); the measured concentrations at Conness Glacier are remarkably consistent and clustered tightly around the origin of the isochron plot. If not erosion, the third explanation is that talus and debris falling off the cirque headwall may have covered the bedrock sites after deglaciation. The consistency of the nuclide measurements is difficult to reconcile with the spatial heterogeneity of a talus field,



and debris burial would have needed to have happened immediately after deglaciation to keep concentrations near blank levels. While this hypothesis cannot be completely rejected, it requires an unlikely set of conditions.

We find the simplest explanation for the extremely low nuclide concentrations to be that Conness Glacier buried our sample sites for the duration of the Holocene. This finding challenges previously published work that projects glacier disappearance in the Sierra Nevada from 10–3 ka prior to 'Neoglacial' advance in the late Holocene (Bowerman and Clark, 2011; Konrad and Clark, 1998; Porter and Denton, 1967). Whether Conness Glacier is representative of the entire Sierra Nevada or not remains an open question. It has retreated more by percentage of its LIA area than the composite of Sierra Nevada glaciers (Basagic and Fountain, 2011, Figure 2) and the most amongst the four glaciers considered here. It therefore could be an outlier in

the Sierras, outpacing other glaciers in its retreat. It is also possible that the hypsometry of Conness Glacier (e.g. headwall shading, orientation, etc.) make this glacier resilient in the early to mid-Holocene while others in the Sierra Nevada disappeared. Further research is needed to assess whether Conness Glacier is representative of the Sierra Nevada or an outlier, and to provide more insight into potential debris burial.

**5.2 Understanding the non-uniform signal**

We seek to understand why four North American glaciers would advance beyond their modern positions many thousands of years apart despite presumed common climate forcing over the Holocene and through the industrial era. While there is certainly spatial climate variability over the Holocene, it seems unlikely that within western North America there would be variation large enough to cause the four glaciers to be thousands of years out of sync. Only Holocene climate records from near

each glacier with well-resolved chronologies could untangle these differences, and such terrestrial archives are lacking. Reviews of Holocene glacier change in western Canada (Menounos et al., 2009) and Holocene temperature and precipitation variability in North America (Shuman and Marsicek, 2016) suggest synchronicity across the region, rather than variability. Similarly, across various spatial scales, from the global temperature reconstructions (Kaufman et al., 2020) to Northern Hemisphere mid-to-high latitudes, to western North America, the trendlines in proxy data remain similar, suggesting commonality across scale (Figure 7).

We find the most parsimonious explanation of the non-uniform bedrock burial durations to be that glaciers advanced roughly in concert across North America, as predicted, but have retreated non-uniformly in the industrial era. Our Holocene exposure-burial histories are predicated upon a comparison to today's glacier positions. Our initial assumption was that the modern position of each glacier would be similar relative to its Holocene length history, but if this is not the case, then modern position is not a common reference point relative to the Holocene. Modern glacier positions reflect the magnitude of modern

retreat, which is occurring at a vastly different timescale than Holocene glacier advance. Glaciers are transiently responding to geologically abrupt warming, where small differences in glacier response time amplify any relative difference between glaciers that might exist over longer timescales. We observe that the amount of retreat relative to LIA extent varies widely amongst the four glaciers we present here (Figure 2). Given this evidence, we consider it more likely that glaciers advanced past their modern sizes thousands of years apart because their modern positions are more variable than in the past, rather than stemming from

disparate Holocene advance histories.

In support of this hypothesis, there appears to be a connection between the amount of retreat relative to LIA extent, glacier response time, and how far each glacier has receded relative to its Holocene advance history. Conness likely has the fastest response time of the four glaciers (smallest area; steepest slope) and thus we expect it would retreat the most relative to its LIA extent and Holocene advance history. This is what we observe: Conness is the most retreated relative to its LIA extent and

the only glacier to have receded past its minimum size of the Holocene. JIF Glacier, on the other hand, is two orders of



magnitude larger, has a lower slope, and has the slowest approximate response time we calculated ($\tau = 27$); we expect it would retreat the least relative to its LIA extent and Holocene advance history. This is also largely what we observe: it has retreated the least relative to its LIA extent and nearly the least relative to its Holocene advance history (it last was a similar size at ~2 ka versus Mammoth Glacier at ~1 ka). Taken together, a simple relationship emerges: the quicker a glacier can respond to
industrial-era climate change, the more it has retreated relative to its LIA extent, and the further back into its Holocene advance history it resides today.

     This relationship holds for Kokanee Glacier, but not for Mammoth Glacier. Kokanee Glacier has retreated 46% from its LIA extent, an intermediate amount between Conness (14%) and JIF (70%) Glaciers. It has the second-steepest slope behind Conness and the quickest response time we calculated ($\tau = 14$), with Conness Glacier's response expected to be quicker given its
steeper slope. We expect then Kokanee would retreat to an intermediate position relative to its Holocene length history, which it did. It has retreated to a size last occupied at ~6 ka; not as far back in time as Conness but more than JIF Glacier at ~2 ka. Mammoth Glacier, though, does not fit the overall pattern. Mammoth Glacier has a low slope and lower response time ($\tau = 23$), closer to JIF Glacier ($\tau = 27$). It has lost nearly the same area from its LIA extent as Kokanee Glacier. We would then expect it to retreat further back relative to its Holocene advance history than JIF Glacier, yet it is the least retreated relative to its Holocene
advance history, last occupying its present-day position at ~1 ka. It is difficult to interpret whether this is an outlier to the trend or proves it false. As noted earlier, Mammoth Glacier's doubling in area within the last millennium is a surprising finding if purely climate driven. It is simpler to explain the rapid advance with non-climatic length changes perhaps related to bed morphology (i.e., the glacier debuttressing from a bedrock knob or spilling out of its cirque and rapidly advancing). If true, then Mammoth Glacier's Holocene length history is not representative of climate and should be excluded. In either case, the complexity
underscores the need to consider glacier hypsometry and ice dynamics when interpretating climate from paleoglaciers.

     Regardless of the strength of the connection between response time and bedrock burial duration, there is clear non-uniformity amongst the four sites relative to their Holocene length fluctuations. We cannot rule out regional climate variability over the Holocene as the source of the variation. However, we find it more convincing that disparate amounts of modern retreat have caused some glaciers to recede further back than others relative to their Holocene advance histories. The pace and intensity
of modern climate change exacerbates normal variation in length response between glaciers due to unique hypsometry and response time. Glaciers in western North America have faced extremely rapid warming, experiencing the coldest decades of the Holocene to the warmest within a little over a century (Marcott et al., 2013). It appears plausible that the rate and magnitude of modern warming has exaggerated any pre-industrial length variability between glaciers that existed across the region. If true, then this implies that modern glaciers are no longer behaving in concert like they did over the Holocene and are now uniquely
out of sync. More well-resolved Holocene paleoclimate records from near active glaciers in western North America would provide clarity on this hypothesis, along with flow-line modeling to compare the regional glacier response to abrupt climate change versus slow climate change. The impact of spatially heterogeneous climate changes, such as arctic amplification or regional precipitation change, can also be investigated for their impact on individual glacier retreat.

**6 Conclusion**

     All four North American glaciers expanded from the early to late Holocene and today are 14-70% of their maximum Holocene extents recorded during the LIA. Our results provide spatial constraints to understand modern glacier position within the context of the Holocene, and support that modern retreat is a reversal of a long-term trend of glacier advance in western



North America. JIF Glacier has been larger than its modern position from at least ~2 ka onwards. Kokanee Glacier has receded to
a position last occupied at ~6 ka. Mammoth Glacier has receded to its size last occupied at ~1 ka and nearly doubled in area over
the last millennium. Conness Glacier has likely receded past its minimum Holocene position. Bedrock erosion rates are on the
low end of published estimates for alpine glaciers. Our results fit best with erosion rates below 0.5 mm yr$^{-1}$, except in instances
where subglacial quarrying is likely to have occurred, and at JIF Glacier where erosion may be higher than we calculate. The
four North American glaciers exhibit surprisingly disparate Holocene exposure-burial histories relative to their positions
occupied today. We argue that this variability is related to the magnitude of modern retreat, rather than asynchronous behavior
over the Holocene, and how our method treats the modern glacier position as a common reference point. In the face of rapid
warming in the industrial era, differences in glacier hypsometry and response time amplify or dampen a glacier's climate
response, growing the relative length differences between glaciers. We hypothesize that the wide range in glacier lengths
observed is a departure from broadly synchronous fluctuations over the Holocene. Comparing modern-day glaciers to a
Holocene baseline is nuanced because glaciers are complex transfer functions of climate that require accounting for glacier
hypsometry and ice dynamics.

**Code availability**

Monte Carlo forward model code used here is available via Vickers et al. (2020).

**Data availability**

All data presented here is available in the main-text tables and supplementary tables. Cosmogenic nuclide measurements will
also be deposited into the ICE-D database ([https://version2.ice-d.org/alpine/](https://version2.ice-d.org/alpine/)) following publication.

**Declaration of competing interest**

The authors declare that they have no competing interests that influenced the research presented in this publication.

**Author contribution**

SAM, JDS, and BMG designed the experiments and acquired funding. AGJ, DLG, TMK, SAM, JDS, BMG, and DHK collected
samples. AGJ, TMK, DLG, SAM, and BMG processed samples. MAC leads isotope measurement lab where samples were
measured. DLG and JDS developed the model code. AGJ and MR made visualizations and mapped glacier areas. AGJ and SAM
wrote the original draft, BM and JDS assisted extensively in manuscript conceptualization, and all authors participated in the
final writing and editing of the paper.

**Acknowledgements**

This research was funded by National Science Foundation Grant EAR-1805620, the Tula Foundation, the National Sciences and
Engineering Research Council of Canada, and Canada Research Chairs. We thank Inyo National Forest, the Bridger-Teton
National Forest, and the Provincial Park Service of Canada. We thank Ben Pelto for helping us access Kokanee Glacier, Aaron
Barth and Alden Laev for field assistance in collecting the Mammoth Glacier samples, and Luke Zoet and Summer Rupper for
thoughtful discussion.



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
