# Peer review of "Four North American glaciers advanced past their modern positions thousands of years apart in the Holocene"

_EGUsphere, 2023_

## Referee Comment (RC1)

**Review of "Four North American glaciers advanced past their modern positions thousands of years apart in the Holocene"**

**Letter**

Dear Editor,

I greatly appreciate the opportunity to review this paper. I also thank the authors for preparing this manuscript.

The research in this manuscript examines the retreat histories of four glaciers in western North America since the start of the Holocene using $^{10}$Be and $^{14}$C and a model of possible retreat and erosion scenerios that could result in these measured values. The results show that the glaciers experience different retreat histories, despite the expectation that they experienced similar climate forcings. Furthermore, the role of hypsometry and glacier response time drive some of the variability between the four glaciers.

I enjoyed reading this manuscript and believe that it could be a contribution to the literature. However, there are several large issues that that I believe need to be resolved or clarified before publication. Some additional interpretion of the data is needed. Additionally, at times, the writing needs improvement, reorganization and clarification. Comments related to the presentation are presented below, however, the list is not extensive.

The matters presented below can largely be addressed, in my opinion. In turn, *major revisions* seem appropriate. My comments are in normal font and quotes from the text are in italics.

I wish the authors the best in developing this work and hope that this review is helpful to that end.

**General comments**

- The authors seem to suggest that variable glacier response times and their current position, as opposed to climate variations, drive the different exposure-advance histories in front of the glaciers. Do more local climate records exist for each glacier that could replace or supplement Figure 7 to verify that the climate was indeed consistent in these areas? This statement should better supported: *Although it is possible to interpret the range of bedrock burial durations as bellwethers of previously unrecognized climate heterogeneity over the Holocene.* For instance in lines 300 to 305, glacier extent and climate is determined from lake sediments for the Sierras, then more global sites are mentioned. Surely there are climate differences across some spatial scales. In the most basic case, better describing the consistent climate will make the paper more accessible to those without a paleoclimate background.

- Some what related to the last point, the study site at Coness Glacier needs special attention. This is a very small glacier and it is likely that ice flow is minimal. As a result, factors such as slope, mass balance gradient, and glacier response time might have limited meaning here. Mauro Fischer and Matthias Huss have several papers on the differences in behavior of small glaciers in a changing climate compared to larger ones. The authors should integrate some of thier work (or similar) into the findings here. I find this especially important given that Coness glacier is an end member in the findings presented here.

- smaller/similar note: is JIF glacier connected to the Juneau Icefield? this will surely impact its response time.

- Some matters in the model need clarification and reanalysis. First, I did not find a convergence test or criterion to show that the number of model runs was adequate to yield a robust result.

It *seems* that erosion occurs only when the rock is covered by ice, which is good, but I am not certain that this is what was done from reading the text.

Furthermore, in my interpretation of the text, erosion rates do not vary through the model run, but are held constant through the model run. There has been much research in recent years to show the substantial temporal and spatial variability in glacier erosion, subglacial sediment accumulation and sediment export (Herman et al., 2015, Lai and Anders, 2021, Seguinot and Delaney 2021, Delaney and Anderson 2022, Stevens et al, 2022, amongst others). In the most basic sense, abrasion predominantly comes from sliding, which itself largely depends on ice thickness, which changes substantially through the study period. As a result, I find that keeping the erosion rate constant through the model run is inappropriate. While the number of needed model runs will substantially increase, I believe that variability in erosion needs to be addressed, especially given the findings presented later. This process would be especially important at Kokanee glacier, where advances and retreats could have occurred.

Also the range of possible erosion rates should be increased past 0.5 mm a-1. This is smaller than the erosion rate of many glacierized catchments, much less below a glacier. This is especially true given the evident impact of erosion the results of JIF glacier, and the potential of erosion to impact the histories of other glaciers.

- Consistency and organization in writing. Writing style is largely a matter for the editor and the authors. However, there are many instances where I found the writing need improvement in consistency, precision and organization. Use of the word "how", for instance, in parts of the abstract and introduction, I found lead to unspecific and vague analysis. I recommend changing throughout. Some specific issues are addressed below. Note that despite these issues, I found several parts well written, and I am confident that with some help and time these matters can be resolved well.

**Specific comments**

- **Ln 20**. Seems like a word is missing and aren't all glaciers either larger or smaller than in the past?

- **Ln 29**. *the ...* variable? spatially changing? ... *intensity and rate of modern warming.*

- **Ln 34**. *glacier change.* volume change, velocity change, length change, slow done? can be more precise.

- **Ln 40**. *How glaciers...* seems unconnected to this paragraph

- **Ln 42**. what is an *extended ice position*?

- **Ln 49–55** some citations are needed. I also recommend considering Anderson (Geology, 2014) and Rowan (ESPL, 2022) to better interpret the moraine records.

- **Ln 58** *how their length incorporates climate* this is quite vague.

- **Ln 62** *industrial change...* in climate? precipitation? pollution?

- **Ln 89–107** much of this is background knowledge (or even results (should a citation not be present)), as opposed to "Glacier setting...'

- **Ln 108** Is JIF glacier connected to the Juneau Icefield? if so this could have some pretty big implications for the response time compared to Coness for instance.

- **Ln 110** Surface slope is brought up here and in other locations through the manuscript. However, given the retreat and advance variability found later, these slopes will have surly changed through the Holocene. Would this impact your analysis?

- **Figure 2** the vertical axis is strange in that it is a percent of area (m$^2$) to extent (I think of as length (m)), so I am not quite sure what it represents or if units are correct. Clarify.

- **Table 1** *trimline and moraines...* This is not my expertise however, I am aware of much debate about the meaning of trimlines, and previous work about moraine records have been mentioned above. Please consider if relevant.

- **Section 3** This section appears to be more about "Methods" as opposed to "Materials"

- **Ln 183** *We assume all samples experienced the same...* all glaciers from the same glacier? please be more precise.

- **Ln 189–196** some of this might be considered a result or source of uncertainty that should be discussed later, in my opinion.

- **Ln 199–200** This sentence might need to be more precise, I am not sure it establishes the precise role of the model.

- **Ln 213–214** *100,0000 scenerios is tested 26 times.* I do not understand this.

- **Ln 214–215** $2-\sigma$ *uncertainty...* of what? the concentration? also is the 4% error in the measurements accounted for in the inversion? if not then it seems that the $2-sigma$ uncertainty is too precise.

- **Figure 4** I like this Figure quite a bit! However, the points outside the envelope are quite interesting to me. Can this analysis be moved up in the results?

- **Figure 5** Sample locations for Kokanee glacier and Mammoth glacier seem to suggest that that the samples we not exposed at the same time. Please comment on this. Such variability might impact the exposure-burial histories in Fig 6.

- **Ln 283** What is the criterion for a sucessfully reproducing the erosion rates?

- **Ln 284** Given that the mean erosion landscapes is on the order of 1-2mm $a^{-1}$ (Hallet et al., 1996), I am not sure that .5 mm $a^{-1}$ is especially high.

- **Figure 6** Since the model is run in with MC, a range of model runs or confidence intervals can be shown on this plot. This would greatly increase the confidence of the results.

- **Ln 295** *all four glaciers ... glacier expansion* How it is known from your data that Conness glacier expanded as it was covered the whole time?

- **Ln 296** *early to late Holocene* isn't this the whole of the Holocene?

- **Ln 300–304** Without knowing the specific papers regarding sediment flux into lakes, it has been well documented in recent research that glacier retreat, as opposed to advance, can lead to increased sediment discharge from glacierized catchments. Thus, please reconsider the robustness of this analysis.

- **Ln 303–305** These regions are far apart with different climate forcings. To make this statement more than correlation, analysis of the global climate is needed. This should be commented upon.

- **Figure 7** Please explain the meaning in the hydroclimate index. What is wetter, for instance?

- **Ln 349–357** To me, this paragraph seems highly speculative, especially as the model does not recreate the rations. Why is it then the preferred explanation? I would be hopeful that more modeling work would be useful in sorting this out.

- **Ln 365** *JIF Glacier is uniquely erosive amongst the four glaciers studied here.* I am not sure how this can be varified. While JIF glacier does plot outside of the envelop in Fig 4, the role of erosion could also impact the histories of the other glaciers, yet the impact could be less suspect because they plot within reasonable values.

- **Ln 375–377** *interspersed episodes of glacier retreat...retreats were of minimal size if they occurred at all.* These sentences seem contradictory. Why would retreats not occur at all, given Fig6. Also, I was tempted to draw another conclusion from this paragraph until I referred to Fig 6. Please reference a figure in this paragraph to support your findings.

- **Ln 378–390** This could be considered results.

- **Ln 389** What simplifications were made in the isochron plot?

- **Ln 391** *Mammoth Glacier is 50%... doubled in size...* I do not follow, maybe it is a problem with the tense.

- **Ln 394** *ice dynamics plays a role...* doesn't ice dynamics always play a role?

- **Ln 398** I think this sentence needs to be more precise.

- **Ln 414** What about subglacial sediment deposition and insulation from erosion? (Beaud et al 2018, Delaney and Anderson 2022, Stevens et al. 2022).

- **Ln 419** *buried.* By ice or by debris?

- **Ln 440–450** I think there is some valuable information in this paragraph. However, much of it is quite general and difficult to link to the paper's findings. Some time distilling this is needed.

- **Ln 449** *their modern positions are more variable that in the past.* I am not sure how this conclusion was arrived at.

- **Ln 459** Please describe a *quicker glacier.*

- **Paragraph 462–475** What is the input data for the glacier response time?

- **Ln 493** *that modern retreat is a reversal of a long-term trend of glacier advance in western North America...* where do the data presented in this paper show that the glaciers have advanced, as opposed to simply covered these bedrock areas? is "modern" considered since the LIA?

---

## Author Response (AR1)

**Reply to Reviewer Comments**

*(Note to the editor: This document was originally posted online on 21 Aug 2023. There may be some extremely minor word choice differences between the changes proposed here and the revised document submitted on 25 Sep 2023. The track changes documents shows all changes from the original submission to the final revised document.)*

Dear Editor,

We thank Reviewer #1 and Reviewer #2 for their thoughtful commentary. Both reviewers clearly spent a lot of time on their reviews and the proposed changes undoubtedly improve our manuscript. Their comments focus on similar aspects of our work: the hypothesis for the non-uniform burial durations, the incorporation of subglacial erosion in the Monte Carlo forward model, and the need for further clarity in writing and organization. Following their suggestions, we propose several changes to the text to address these three areas. The changes are summarized below:

**Hypothesis for non-uniform burial duration**
- We are more conservative about our hypothesis for the non-uniform burial durations throughout the manuscript.
- We remove the hypothesis about glaciers today being more 'out of sync' than other times in the Holocene on the grounds that we cannot rule out regional climate variation over the Holocene as a driver of the non-uniform burial durations.
- We moderate the Discussion section (5.2) on understanding the non-uniform signal to be more cautious in our interpretations.

**Monte Carlo forward model and subglacial erosion**
- We rewrite the Methods section (3.3) that describes the Monte Carlo forward model to provide greater clarity.
- We report in the Results the number of overlapping Holocene exposure-burial scenarios found for each glacier by our Monte Carlo forward modeling. We also report the number of plausible histories found for each sample in the supplement (Figure S5).
- We present mean abrasion rates at three of the four glaciers from our Monte Carlo forward model results. We add a new figure showing these histograms in the supplement (new Figure S7)
- We perform an additional modeling exercise (new Figure 8) to better understand the $^{14}$C-$^{10}$Be ratios at JIF Glacier and constrain erosion rates. We revise the paragraphs interpreting the Holocene history of JIF Glacier (Lines 335–367). We note that our conclusion of burial at ~2 ka remains the same.

**Organization and clarity in writing**
- We reorganize key aspects of the manuscript: we move the mapping, hypsometry, and response time calculations from the Background section to Methods and Results, and we reorganize the Discussion to begin with the 'simple' glaciers (Kokanee and Mammoth) and get progressively more complex (Juneau Icefield, Conness)
- We break the Results into three sections: 4.1 $^{14}$C-$^{10}$Be ratios, burial isochrons, and exposure ages, 4.2 Monte Carlo forward model results, and 4.3 Mapping, glacier hypsometry data, and response time calculations.
- We provide more detailed descriptions of our references throughout the manuscript, especially in the Introduction and Background, leading to a more streamlined Introduction/Background.
- We move information in figure and table captions into the body of the main text.

We address the comments in order, starting with Reviewer #1. Reviewer comments are presented in their original format and our response is in blue text, with **manuscript changes in blue and bold.** To streamline review, for small changes on grammar, word choice, etc. we write 'Done' to signify we have made the change. When large sections have been revised, we copy/paste the new text and refer to it. We do not bold-face these paragraphs because it makes them hard to read.

**Reviewer #1**

I greatly appreciate the opportunity to review this paper. I also thank the authors for preparing this manuscript.

The research in this manuscript examines the retreat histories of four glaciers in western North America since the start of the Holocene using 10Be and 14C and a model of possible retreat and erosion scenerios that could result in these measured values. The results show that the glaciers experience different retreat histories, despite the expectation that they experienced similar climate forcings. Furthermore, the role of hypsometry and glacier response time drive some of the variability between the four glaciers.

I enjoyed reading this manuscript and believe that it could be a contribution to the literature. However, there are several large issues that that I believe need to be resolved or clarified before publication. Some additional interpretion of the data is needed. Additionally, at times, the writing needs improvement, reorganization and clarification. Comments related to the presentation are presented below, however, the list is not extensive.

The matters presented below can largely be addressed, in my opinion. In turn, major revisions seem appropriate. My comments are in normal font and quotes from the text are in italics.

I wish the authors the best in developing this work and hope that this review is helpful to that end.

**General comments**

- The authors seem to suggest that variable glacier response times and their current position, as opposed to climate variations, drive the different exposure-advance histories in front of the glaciers. Do more local climate records exist for each glacier that could replace or supplement Figure 7 to verify that the climate was indeed consistent in these areas? This statement should better supported: 'Although it is possible to interpret the range of bedrock burial durations as bellwethers of previously unrecognized climate heterogeneity over the Holocene.' For instance in lines 300 to 305, glacier extent and climate is determined from lake sediments for the Sierras, then more global sites are mentioned. Surely there are climate differences across some spatial scales. In the most basic case, better describing the consistent climate will make the paper more accessible to those without a paleoclimate background.

Reviewer #1 asks about climate differences across spatial scales during the Holocene. First, to address the reviewer's comment (along with a similar point by Reviewer #2), we moderate our language throughout the manuscript about whether spatial variability over the Holocene or retreat since the industrial era is the main driver of our finding of non-uniform burial durations. As both reviewers note, we cannot rule out

local climate heterogeneity in the Holocene unless we have local data to suggest otherwise. Second, we may have caused confusion with our language about heterogeneous climate itself versus heterogeneous climate *change*. The point we are trying to make is that the trends in Holocene climate *change* across western North America seem to be broadly synchronous, which matters for our purposes because we are studying glacier length change over the Holocene. To clarify this, **we modify Line 437 to: "…suggest synchronous change across the region, rather than variable change…"**

Regarding finding local records, we cite the paleoclimate data from Routson et al., 2021 ("A multiproxy database of western North American Holocene paleoclimate records") as it is the most complete Holocene climate analysis for western North America to date. Ideally, we would present a robust paleoclimate record from near each glacier. We searched the Routson et al., 2021 dataset, however, and could not find satisfactory records at all four sites. It was not clear what distance would satisfy as sufficiently 'local' and temporal coverage was quite varied (not many long records and often at low age resolution). In the absence of ideal records, we rely on the arguments made in Menounos et al. 2009 and Shuman and Marsicek, 2016 about how trends in Holocene climate change across western North America appear to be similar through the Holocene. To make this logic clear to the reader, **we change Line 434 to: "Sufficient local terrestrial archives are lacking to determine if spatial climate variability is influencing the differences in our observed glacier changes (Rouston et al. 2021). Therefore, we turn to the reviews of Holocene glacier change in western Canada (Menounos et al., 2009) and Holocene temperature and precipitation variability in North America (Shuman and Marsicek, 2016) that suggest synchronicity across the region, rather than variability."**

We also present the geologic records of glacier change from moraines and distal lake sediment fluxes at each site in more detail to provide the best-possible insight into local glacier change. This was suggested by Reviewer #2. As such, **much of Section 2 Background (Lines 89–147 of original manuscript) has been revised. It now reads:**

**"2 Background**
The four North American glaciers in this study are located along the American Cordillera between 38–60° N (Figure 1, inset map). From north to south, the first glacier is an unnamed valley glacier in the Juneau Icefield that we henceforth refer to as JIF Glacier (59.47° N, 135.96° W, 1492 m). JIF Glacier is located in the Coast Mountains on the border of southeast Alaska and British Columbia. It is an independent glacier today within the broader Juneau Icefield due to a drainage divide at its headwall. The Holocene history of Alaskan glaciers is thought to follow Northern Hemisphere summer insolation, with minimum glacier positions in the early Holocene and maximum glacier positions in the late Holocene (1810–1880 CE) as insolation decreases from ~9 ka to present (Barclay et al. 2009). Overridden tree stumps and detrital wood suggest that two land-terminating outlet glaciers of the Juneau Icefield advanced at ~2 ka past their 'modern' (at time of field work in 2004 CE) position (Clague et al., 2010). In the Chugach Mountains, ~600 km northwest of the Juneau Icefield, distal lake sediment records suggest glaciers disappeared there from 10–6 ka before the onset of Neoglaciation at ~4.5 ka (McKay and Kaufman, 2009). The authors document synchronous glacier advances at their two Chugach sites in the last two millennium, one at ~2 ka, and the other during the LIA (1400–1900 CE, or 0.6–0.1 ka), in agreement with the late-Holocene advances observed in the Juneau Icefield (Clague et al., 2010). Distal lake sediments from the Ahklun Mountains of southwestern Alaska also suggest that glaciers disappeared locally from 9–3 ka before Neoglaciation at ~3 ka (Levy et al., 2004). In sum, the early and mid-Holocene history of glaciers in southern Alaska appears to be spatially variable but there is consistent, repeated evidence of late Holocene glacier growth.

Kokanee Glacier (49.75° N, 117.14° W, 2561 m) is in southeastern British Columbia in the Selkirk Mountains. A review of Holocene glacier fluctuations in the Canadian Cordillera using lake sediment records, moraine dendrochronology, and lichen ages (Menounos et al., 2009) finds that from 11–7 ka, ice was smaller than its extent in the late 20th century. Episodic glacier advances occurred at

8.6–8.1 ka, 7.4–6.5 ka, 5.8 ka, 4.4–4.0 ka, 3.7–2.8 ka, 1.7–1.3 ka, and through the LIA (Menounos et al., 2009; Osborn et al., 2012). Synchronous Holocene advances occurred amongst the southern Canadian Cordillera on the centennial scale, suggesting that these advances are related to broad climate signals across western North America (Menounos et al., 2009).

Mammoth Glacier (43.17°N, 109.67°W, 3627 m) is in the Wind River Range, Wyoming. Its Holocene length history is expected to follow that of the western United States and Canada: glaciers retreated during deglaciation and reached Holocene minimum positions between 11–7 ka, then advanced to maximum extents during the LIA (Marcott et al., 2019; Menounos et al., 2009). Distal lake sediment fluxes from the valley below Mammoth Glacier suggest that the glacier has been active since at least 4.5 ka—though perhaps disappearing in the early Holocene—but was much smaller than its LIA extent from 4.5–1 ka (Davies, 2011), implying a substantial advance during the LIA.

Conness Glacier (37.97°N, 119.32°W, 3603 m) is in the Sierra Nevada, California, USA. Glaciers in the Sierra Nevada are thought to have disappeared entirely by the early Holocene (~10 ka) prior to so-called 'Neoglacial' advances beginning at ~3 ka (Bowerman and Clark, 2011; Cary, 2018; Porter and Denton, 1967). Progressively increasing fluxes of rock flour from ~3 ka to the LIA have been documented in distal lake sediments at Conness Glacier (Konrad and Clark, 1998) and distal lake sediments at Palisade Glacier (Bowerman and Clark, 2011, located ~125 km south of Conness Glacier), interpreted as the reformation and advance of both glaciers.

**Because we remove the mapping, hypsometry, and response time calculations from Section 2, below we copy the revised sections of Methods (3.4) and Results (4.3):**

**"3.4 Glacier mapping, hypsometry, and response time**

Although we are primarily concerned with length fluctuations of the four glaciers in the Holocene, our measurements are inherently linked to the modern-day position of each glacier because it is the reference point. It is therefore necessary to understand the history of each glacier's position since the industrial era (and associated glacier dynamics) that have influenced its modern position. We first mapped modern (2021 CE) glacier area against LIA glacier extents in QGIS to characterize glacier retreat over this period (Figure 1). We inferred the LIA glacier extents from moraines and trimlines. We overlayed glacier outlines onto modern satellite imagery from Copernicus. We also included an intermediate position using the outlines available from the Randolph Glacier Inventory (RGI Consortium, 2017). The RGI data are from miscellaneous years based on what was available in RGI. The amount of retreat since ~1880 CE is related to the climate change at the glacier and the glacier's response time, defined as the time for each glacier's length to reach equilibrium with climate change. Hypsometric variables such as ice surface slope, cirque-wall shading, and debris cover impact response time, with steep ice surface slopes thought to be a particular correlate for quick response times (Pelto and Hedlund, 2001; Zekollari et al., 2020). We approximate each glacier's response time according to Jóhannesson et al. (1989), where mean glacier thickness is divided by maximum mass loss from the terminus as shown in Eq. 2:

$$\tau = \frac{H}{-b_t} \tag{2}$$

where $\tau$ = time for volume adjustment in years, $H$ = thickness in meters (m), and $b_t$ = maximum mass loss at the terminus in m yr$^{-1}$. Mean glacier thickness is taken from Farinotti et al. (2019), and maximum mass loss at the terminus is from Hugonnet et al. (2021). This approximation for response time (along with glacier slope) provides insight into the glacier's response to industrial-era warming. We note this calculation is a minimum estimate of how quickly a glacier could adjust its volume because we use the maximum mass loss from the terminus; the purpose of this calculation is merely to serve as a common point of comparison amongst the four glaciers, rather than a robust estimate of glacier response time. We present glacier areas in Figure 6 and report all glacier hypsometry data and response time calculation details in Table 2.
…

**4.3 Glacier mapping, hypsometry, and response time calculations**

The glaciers retreated 14–70% from their maximum Holocene extents recorded during the LIA (Figure 6). Basagic and Fountain (2011) mapped the industrial-era retreat for the Sierra Nevada, and Devisser and Fountain (2015) for

the Wind River Range. We included their intermediate extents from historical photography at Mammoth and Conness Glaciers. We also included their composite data from glaciers in the Sierra Nevada and the Wind River Range to provide insight into how each glacier compares to the rest of the glaciers in its region. Conness Glacier has retreated more quickly than the composite of glaciers for the Sierra Nevada, while Mammoth Glacier's retreat closely matches the average retreat for glaciers in the Wind River Range.

[Figure]

**Figure 6.** Glacier area through the industrial era relative to glacier area at the end of the Little Ice Age (LIA). LIA moraines assumed to be last occupied at ~1880 CE (Menounos et al., 2009; Wanner et al., 2008). Two composites of regional glaciers from the Wind River Range and Sierra Nevada are included to provide context for regional glacier change (DeVisser and Fountain, 2015; Basagic and Fountain, 2011). Internal mapping has been supplemented with mapping by DeVisser and Fountain (2015) at Mammoth Glacier and Basagic and Fountain (2011) at Conness Glacier. See Table S5 for calculation and source details.

Glacier hypsometry data and response time calculations are presented in Table 2. The four glaciers can be subdivided into two groups. JIF Glacier and Mammoth Glacier are characterized by their shallow slopes and slow response times. JIF Glacier has the shallowest ice surface (5°) and the longest response time ($\tau = 27$ yrs) of the four glaciers. It is also the largest glacier studied here, with an area of 15.2 km$^2$ in 2021 (compared to the smallest glacier presented, Conness Glacier, at 0.1 km$^2$). Mammoth Glacier has the second-shallowest ice-surface slope at 9° and the second-slowest response time ($\tau = 23$ yrs). Kokanee and Conness Glacier are characterized by their relatively steep slopes and quick response times. Kokanee Glacier has a steep ice-surface slope at 22° and the second quickest response time ($\tau = 14$ yrs). Conness has the steepest ice surface slope, 23°, suggesting the quickest response time of the glaciers studied here given the relationship proposed by Zekollari et al. (2020). We omit Conness Glacier from these calculations because there was a large discrepancy between the area of the glacier today and the area of the glacier used in the mean thickness calculations by Farinotti et al. Conness is the smallest glacier in this study with an area of 0.1 km$^2$ in 2021, representing 14% of its LIA maximum extent (0.6 km$^2$). Kokanee Glacier today occupies 46% of its LIA area. Its Mammoth Glacier is in the Wind River Range, Wyoming and is 49% of its LIA maximum area, which is similar to the average of glaciers from the Wind River Range (Figure 6; DeVisser and Fountain, 2015)."

- Some what related to the last point, the study site at Coness Glacier needs special attention. This is a very small glacier and it is likely that ice flow is minimal. As a result, factors such as slope, mass balance gradient, and glacier response time might have limited meaning here. Mauro Fischer and Matthias Huss have several papers on the differences in behavior of small glaciers in a changing climate compared to larger ones. The authors should integrate some of thier work (or similar) into the findings here. I find this especially important given that Coness glacier is an end member in the findings presented here.

Reviewer #1 asks if Conness Glacier behaves differently from larger glaciers such that it may not be comparable to our other sites. In a paper by Huss and Fischer from 2016 titled "Sensitivity of Very Small Glaciers in the Swiss Alps to Future Climate Change", Huss and Fischer conclude that "The mass balance sensitivity of very small glaciers to temperature and precipitation change is similar to that of larger ice masses. However, it is characterized by a strong variability among individual glaciers." **We better clarify the sensitivity of small glaciers to changes in climate by adding a sentence on Line 427 of the original manuscript: "The sensitivity of small glaciers like Conness to changes in temperature and precipitation has been shown to be similar to that of large glaciers, but the sensitivity has also been shown to be more variable from glacier-to-glacier than for large glaciers (Huss and Fischer, 2016)."** We note that findings from Zekollari, Huss, et al., 2016 were used in our study and critical to developing our hypothesis about response time.

- smaller/similar note: is JIF glacier connected to the Juneau Icefield? this will surely impact its response time.

Our treatment of JIF Glacier as an independent glacier is based on a drainage divide at its headwall. Our mapping also uses the polygon from the Randolph Glacier Inventory (2017), which is derived from strict rules that delineate individual glaciers from larger ice mass complexes. **To clarify this, we add a sentence on Line 110 of the original manuscript referring to JIF Glacier: "It is an independent glacier today within the broader Juneau Icefield due to a drainage divide at its headwall."**

- Some matters in the model need clarification and reanalysis. First, I did not find a convergence test or criterion to show that the number of model runs was adequate to yield a robust result. It seems that erosion occurs only when the rock is covered by ice, which is good, but I am not certain that this is what was done from reading the text.

  Furthermore, in my interpretation of the text, erosion rates do not vary through the model run, but are held constant through the model run. There has been much research in recent years to show the substantial temporal and spatial variability in glacier erosion, subglacial sediment accumulation and sediment export (Herman et al., 2015, Lai and Anders, 2021, Seguinot and Delaney 2021, Delaney and Anderson 2022, Stevens et al, 2022, amongst others). In the most basic sense, abrasion predominantly comes from sliding, which itself largely depends on ice thickness, which changes substantially through the study period. As a result, I find that keeping the erosion rate constant through the model run is inappropriate. While the number of needed model runs will substantially increase, I believe that variability in erosion needs to be addressed, especially given the findings presented later. This process would be especially important at Kokanee glacier, where advances and retreats could have occurred.

We agree with both reviewers that properly accounting for subglacial erosion is critical for understanding $^{14}C$-$^{10}Be$ measurements in proglacial bedrock. To address these concerns, we rewrite Methods Section 3.3 which details how the Monte Carlo forward model operates. Including Reviewer #2's comments, it is clear that we did not communicate some of the key aspects of our model. This was in part because it was published elsewhere (Vickers et al., 2020, Geology) and we did not want to repeat text. However, we agree with the reviewers that this manuscript will certainly benefit from greater detail which we now provide. Certainly, a single erosion rate is unlikely to have occurred both spatially and temporally at the bedrock sites in this study. Our Monte Carlo forward model approaches erosion iteratively, testing numerous erosion rates at each sample under an exhaustive list of exposure-burial scenarios to identify all combinations that reproduce the measured concentrations of $^{14}C$ and $^{10}Be$ within measurement and production rate uncertainties (2σ). It then uses these scenarios to provide exposure, burial, and erosion data for the sample sites. **We think that clarifying the text will address reviewer concerns, and we revise Methods Section 3.3 to this end:**

[revised manuscript text omitted]

**Regarding the number of scenarios tested and model convergence, we make three key changes to the Results section on Lines 274–282:**

- First, **we report the number of overlapping solutions from each glacier in Results 4.2** to provide insight into the number of model runs and the robustness of our results. We also report the number of plausible exposure-burial scenarios found for each sample in Figure S5 (attached below, the full Monte Carlo forward model results from each glacier).

- Second, **we more clearly explain that erosion is calculated at each sample and direct readers to the supplement for details.**

- Third, **we report mean erosion rates from each glacier** to make clear that we have thoroughly considered erosion and present the histograms of this data in the supplement (new Figure S7, below).

**The updated Results 4.2 text is copied below (revised version of Lines 274–282):**

**"4.2 Holocene exposure-burial histories and erosion rates from Monte Carlo forward modeling**
We present the results from the Monte Carlo forward model experiments separately from the nuclide ratios alone so the two methods can be compared. We plotted the overlapping exposure-burial scenarios as bedrock 'probability of exposure' through the Holocene (Figure 5). The full model output with plausible exposure-burial histories for each sample and a histogram of the erosion rates used at each sample can be found in Figures S5 and S6 of the supplement.

The Monte Carlo forward model results predict that JIF, Kokanee, and Mammoth glaciers were smaller than their modern size in the early-to-mid-Holocene and larger than their modern size in the late Holocene (Figure 5). At Mammoth Glacier, bedrock shows the most exposure of the four sites (from 11–1 ka) and the latest burial in the Holocene (~1 ka). The number of overlapping scenarios is 124. The probability of exposure is relatively high (80–90%) during inferred exposure, suggesting good agreement amongst the overlapping scenarios. Kokanee Glacier exhibits exposure before ~6 ka and burial after ~6 ka. The number of overlapping scenarios is 137. The overlapping scenarios for Kokanee's bedrock similarly agree well with each other: the probability of exposure is >90% in the early Holocene and <10% by 5 ka. At JIF Glacier, model results suggest bedrock burial at ~2 ka. The number of overlapping scenarios is 282. However, the probability of exposure at any one time is relatively low, never rising above 70%.

We calculated the mean erosion rate at each glacier by averaging the erosion rates used at each sample in the overlapping scenarios and plotted the data as histograms in Figure S7. The mean erosion rate at Mammoth Glacier is $0.7 \pm 0.7$ mm yr$^{-1}$ ($0.2 \pm 0.2$ mm yr$^{-1}$ when a low-concentration sample is excluded, see Discussion); at Kokanee Glacier is $0.04 \pm 0.03$ mm yr$^{-1}$; and at JIF Glacier is $0.3 \pm 0.3$ mm yr$^{-1}$. At JIF Glacier, four samples above the continuous exposure curve in Figure 4 are excluded from these calculations; see Discussion."

**We note that the Monte Carlo forward model is intentionally broad**—it iterates through millions of potential burial, exposure, and erosional scenarios to provide insight into the Holocene history of the glaciers relative to their positions today. We prefer our Monte Carlo randomization approach because it minimizes any inferences of glacier dynamics and Holocene history. For example, if we were to try and alter erosion rates based on ice thickness, we would need to vary ice thickness according to some condition, likely climate, which we worry would impose a circular logic. On the other hand, if ice thickness were randomly varied, it is not clear that this would more accurately represent 'real' glacier conditions than what we have done, which is test millions of iterations and see what can plausibly recreate our measurements. We argue that by sampling erosion iteratively and then selecting the solutions that best fit the data, we are most reasonably approximating 'real' erosion rates.

**Below is the updated Figure S5 of Monte Carlo Forward Model results:**

[Figure]

**"Figure S5.** Full Monte Carlo forward model results of possible exposure-burial histories from each sample at each glacier. The plausible exposure-burial scenarios are plotted as horizontal sequences of yellow and blue timesteps, where yellow represents exposure and blue represents burial (see Figure S5 for illustration). For example, sample KG-04 has 1,247 plausible exposure-burial histories; the scenarios depicted mostly show exposure in the early Holocene and burial in the late Holocene with some variation in individual scenarios. An exposure-burial scenario

can be successful at multiple erosion rates, thus some n-values are > 100,000. The final 'Probability of exposure' is an average of the overlapping scenarios, shown as 'Overlap.' Only scenarios with burial in the final two centuries are included in the overlapping scenarios given geologic evidence for burial prior to sampling."

**Below is the mean erosion rate histograms (new Figure S7):**

[Figure]

**"Figure S7.** Histograms of erosion rates used in overlapping scenarios for all samples at JIF, Kokanee, and Mammoth Glacier. The data in this plot are used to make the mean erosion rate calculations for each glacier. This figure differs from Figure S6 in that Figure S6 shows the erosion rates capable of recreating measured concentrations at each sample regardless of whether the scenario is overlapping. Here, we show only erosion rates used in the overlapping scenarios (successful scenarios for all samples). Sample MG-04 is a relatively low-concentration sample with a higher inferred erosion rate, potentially by subglacial quarrying.

Also the range of possible erosion rates should be increased past 0.5 mm a-1. This is smaller than the erosion rate of many glacierized catchments, much less below a glacier. This is especially true given the evident impact of erosion the results of JIF glacier, and the potential of erosion to impact the histories of other glaciers.

We now try to be as transparent as possible about why we ran two experiments at Conness Glacier. The original experiment tests erosion rates from 0–2.5 mm yr$^{-1}$ like at the other glaciers. We then ran an additional experiment where erosion is capped at 0.5 mm yr$^{-1}$. **We have rewritten the paragraph on lines 283–287 of the original manuscript and it now reads:**

"Nuclide concentrations in the Conness Glacier samples are uniquely low amongst the four glaciers, which poses challenges for our Monte Carlo analysis. At such low concentrations, small differences in blank corrections, scaling schemes, and production pathways have an outsize influence on modeling nuclide concentrations. We initially ran the model using the same parameterizations at the other sites where erosion rates are tested up to 2.5 mm yr$^{-1}$. The model finds 4,914 overlapping solutions and erosion rates skew heavily towards the highest possible rates (2.5 mm yr$^{-1}$; Figure S8). However, erosion rates at the other three sites all skew towards lower rates (< 0.5 mm yr$^{-1}$; Figure S6), and there is no geologic evidence that Conness Glacier is a uniquely erosive glacier. We then re-ran the Monte Carlo forward model with erosion rates capped at 0.5 mm yr$^{-1}$. Under this low-erosion parameterization, no scenarios could reproduce the measured concentrations at all samples (zero overlapping scenarios, Figure S8). Because of these complexities, we do not present modeling results from Conness Glacier in Figure 5 and explore the various ways to interpret the low concentrations in the Discussion."

- Consistency and organization in writing. Writing style is largely a matter for the editor and the authors. However, there are many instances where I found the writing need improvement in consistency, precision and organization. Use of the word "how", for instance, in parts of the abstract and introduction, I found lead to unspecific and vague analysis. I recommend changing throughout. Some specific issues are addressed below. Note that despite these issues, I found several parts well written, and I am confident that with some help and time these matters can be resolved well.

We make several specific changes to the texts enumerated in the line comments below.

**Specific comments**

- Ln 20. Seems like a word is missing and aren't all glaciers either larger or smaller than in the past?
  - Our method directly compares Holocene ice positions to ice positions at the time of sampling (2018–2020 CE).
- Ln 29. the ... variable? spatially changing? ... intensity and rate of modern warming.
  - **This sentence on Line 29 is deleted as part of moderating the language around our hypothesis for non-uniform burial durations.**
- Ln 34. glacier change. volume change, velocity change, length change, slow done? can be more precise.
  - **This sentence on Line 34 is deleted as part of moderating the language around our hypothesis for non-uniform burial durations.**
- Ln 40. How glaciers. . . seems unconnected to this paragraph
  - **Changed Ln 40 of original manuscript to "Glacier length fluctuations in the Holocene provide…"**

- Ln 42. what is an extended ice position?

  - In original manuscript, "expanded" is written.

- Ln 49–55 some citations are needed. I also recommend considering Anderson (Geology, 2014) and Rowan (ESPL, 2022) to better interpret the moraine records.

  - Done.

- Ln 58 how their length incorporates climate this is quite vague.

  - This sentence has been deleted.

- Ln 62 industrial change. . . in climate? precipitation? pollution?

  - **Changed in Ln 62 of original manuscript to** "...recording industrial era **climate changes**..."

- Ln 89–107 much of this is background knowledge (or even results (should a citation not be present)), as opposed to "Glacier setting...'

  - **Moved to methods/results.** See reply to general comments where updated Section 2, 3.4, and 4.3 are copied.

- Ln 108 Is JIF glacier connected to the Juneau Icefield? if so this could have some pretty big implications for the response time compared to Coness for instance.

  - As noted above, we add a sentence on Line 110 of the original manuscript: "JIF Glacier is an independent glacier today within the broader Juneau Icefield due to a drainage divide at its headwall."

- Ln 110 Surface slope is brought up here and in other locations through the manuscript. However, given the retreat and advance variability found later, these slopes will have surly changed through the Holocene. Would this impact your analysis?

  - We only bring in surface slope to aid in discussions of glacier response time during the industrial era. Over the Holocene, we're concerned with millennial-scale changes in glacier position, where response time is less of a consideration. We do not think this would impact our analysis.

- Figure 2 the vertical axis is strange in that it is a percent of area ($m^2$) to extent (I think of as length (m)), so I am not quite sure what it represents or if units are correct. Clarify.

  - **Changed Y-axis label to read "Glacier Area Relative to Little Ice Age Area (%)"**

- Table 1 trimline and moraines... This is not my expertise however, I am aware of much debate about the meaning of trimlines, and previous work about moraine records have been mentioned above. Please consider if relevant.

  - We use trimlines and moraines to infer the Little Ice Age area of the glacier. In an ideal world, there would be written records or photographs of each glacier in contact with its moraine, like some glacier histories in Europe. In the absence of such records in western North America, we use the morphology visible from satellite photography to obtain Little Ice Age area. It is dominantly the moraine that we used to make our maps, and trimline is only used at JIF Glacier where a lateral moraine is difficult to see.

- Section 3 This section appears to be more about "Methods" as opposed to "Materials"

  - **Changed Section 3 header to "Methods"**

- Ln 183 We assume all samples experienced the same... all glaciers from the same glacier? please be more precise.

  - **Changed to** "...all samples **at a given glacier** experienced the same..."

- Ln 189–196 some of this might be considered a result or source of uncertainty that should be discussed later, in my opinion.

  - **Lines 192–196 are deleted for redundancy**; mentioned in the discussion.

- Ln 199–200 This sentence might need to be more precise, I am not sure it establishes the precise role of the model.

  - Done. We have rewritten this section of the manuscript (Methods 3.3), see above reply to general comments.

- Ln 213–214 100,0000 scenerios is tested 26 times. I do not understand this.

  - We have rewritten this section of the manuscript (Methods 3.3), see reply to general comments. Addressing this specific comment, our model uses each paired isotope measurement and attempts to recreate the measured concentration 2.6 million times (100,000 unique exposure-burial scenarios tested at erosion rates of 0.0 to 2.5 mm yr$^{-1}$ in increments of 0.1 mm yr$^{-1}$; i.e. 26 x 100,000). Through testing this large range of potential exposure and erosion histories, and then only accepting scenarios that work for all samples together at a given glacier, we argue that are reasonably sampling the suite of plausible erosion rates.

- Ln 214–215 2–σ uncertainty... of what? the concentration? also is the 4% error in the measurements accounted for in the inversion? if not then it seems that the 2 – sigma uncertainty is too precise.

  - We have rewritten this section of the manuscript (Methods 3.3), see reply to general comments. The Monte Carlo forward model takes into account both measurement uncertainty and production rate uncertainty at 2σ.

- Figure 4 I like this Figure quite a bit! However, the points outside the envelope are quite interesting to me. Can this analysis be moved up in the results?

  - We have revised and rewritten our results and interpretation regarding these data points. Results 4.1 presents the finding, and revised Discussion 5.1 dives into these samples in detail (copied in reply to Reviewer #2's general comments). In short, we perform a new modeling exercise and consider these samples outliers.

- Figure 5 Sample locations for Kokanee glacier and Mammoth glacier seem to suggest that that the samples we not exposed at the same time. Please comment on this. Such variability might impact the exposure-burial histories in Fig 6.

  - We find the consistency in the data at Mammoth and Kokanee to provide good evidence that the sites were buried and exposed contemporaneously, or at least within uncertainty of cosmogenic nuclide dating. The Monte Carlo forward model results match the predictions of the isochron plot, and in both cases the exposure-burial scenarios are well-constrained by the measurements. At JIF Glacier, however, we now raise this concern as a possibility. Please see our rewritten Discussion 5.1 on JIF Glacier in our reply to Reviewer #2's general comments.

- Ln 283 What is the criterion for a sucessfully reproducing the erosion rates?

- - We revise Methods 3.3 to more clearly address this (see above reply to the general comments)

- Ln 284 Given that the mean erosion landscapes is on the order of 1-2mm $a^{-1}$ (Hallet et al., 1996), I am not sure that .5 mm $a^{-1}$ is especially high.

  - Reviewer #2 has a similar point. **We remove 'exceptionally high' in Line 284 from the text.** Lines 283–287 have been rewritten and are found above in the reply to the general comments.

- Figure 6 Since the model is run in with MC, a range of model runs or confidence intervals can be shown on this plot. This would greatly increase the confidence of the results.

  - We now report the number of overlapping exposure-burial scenarios in Results 4.2 Monte Carlo forward model results (see reply to general comments above)

- Ln 295 all four glaciers … glacier expansion How it is known from your data that Conness glacier expanded as it was covered the whole time?

  - **We have changed Line 295 to:** "In agreement with broad evidence of Holocene glacier advance in western North America (Menounos et al., 2009; Solomina et al., 2015; Davis et al., 2009), our results at **three of four sites** suggest glacier expansion from early to late Holocene."

- Ln 296 early to late Holocene isn't this the whole of the Holocene?

  - At the start of the Holocene at 11.7 ka, there is evidence many glaciers in western North America were outbound of their LIA maximum positions (Marcott et al., 2019). The early Holocene is a range of time from ~11–7 ka which we prefer to use because it captures the likely minimum position of glaciers in western North America (Menounos et al., 2009; Solomina et al., 2015) without being too precise about exactly when glaciers were at their smallest position, which we contend remains unknown.

- Ln 300–304 Without knowing the specific papers regarding sediment flux into lakes, it has been well documented in recent research that glacier retreat, as opposed to advance, can lead to increased sediment discharge from glacierized catchments. Thus, please reconsider the robustness of this analysis.

  - We are citing the literature as it exists for this region, though we acknowledge there is debate about lake sediment fluxes. **We have moderated this language and changed Line 300 to: "At Conness Glacier, ice growth cannot be inferred due to near-blank nuclide concentrations, although it seems likely given the evidence for increasing rock flour in the Sierra Nevada in the late Holocene (Konrad and Clark, 1998; Bowerman and Clark, 2011)."**

- Ln 303–305 These regions are far apart with different climate forcings. To make this statement more than correlation, analysis of the global climate is needed. This should be commented upon.

  - We moderate our language about heterogeneous climate versus modern retreat as the primary driver of the non-uniform burial durations (see general comments above).

- Figure 7 Please explain the meaning in the hydroclimate index. What is wetter, for instance?

- **We add a sentence to Ln 331 of the original manuscript stating: "Temperature and hydroclimate indexes are relative to pre-Industrial values, with more positive values being warmer and wetter; negative values colder and drier."**

- Ln 349–357 To me, this paragraph seems highly speculative, especially as the model does not recreate the rations. Why is it then the preferred explanation? I would be hopeful that more modeling work would be useful in sorting this out.

  - We agree with the reviewer. We present new modeling (Figure 8) to address this point which echoes Reviewer #2's call for similar changes. We rewrite Lines 335–367 and we add Figure 8. The new text and the new figure can be found in the reply to general comments from Reviewer #2.

- Ln 365 JIF Glacier is uniquely erosive amongst the four glaciers studied here. I am not sure how this can be varified. While JIF glacier does plot outside of the envelop in Fig 4, the role of erosion could also impact the histories of the other glaciers, yet the impact could be less suspect because they plot within reasonable values.

  - This line has been removed; this section changed after we did the additional modeling exercise for JIF Glacier (new Figure 8, copied in reply to Reviewer #2).

- Ln 375–377 interspersed episodes of glacier retreat…retreats were of minimal size if they occurred at all. These sentences seem contradictory. Why would retreats not occur at all, given Fig6. Also, I was tempted to draw another conclusion from this paragraph until I referred to Fig 6. Please reference a figure in this paragraph to support your findings.

  - **This sentence is clarified. We change Ln 375 to read: "Our results suggest the glacier was expanded beyond its position today from ~6 ka through the LIA, and that any glacier length fluctuations happened between the position occupied at ~6 ka and its LIA maximum extent."**

- Ln 378–390 This could be considered results.

  - **Done. We now present the mean [10]Be exposure ages in Results at each glacier.**

- Ln 389 What simplifications were made in the isochron plot?

  - **Reviewer #2 had a similar comment and we have removed this line.**

- Ln 391 Mammoth Glacier is 50%… doubled in size…I do not follow, maybe it is a problem with the tense.

  - **To address a few comments on this paragraph, we rewrite Lines 378–398 in 5.1 discussing Mammoth Glacier. Lines 378–398 now read as:**

"We interpret that Mammoth Glacier was smaller than its modern area from ~11–1 ka and larger from ~1 ka to the LIA. The high apparent exposure ages of both nuclides in the Mammoth Glacier samples require the glacier to be smaller than its modern area for most of the Holocene. The Monte Carlo forward model results quantify this interpretation, predicting Mammoth Glacier was smaller than its modern area from 11–1 ka before advancing beyond its modern position in the last millennium. An interesting implication of Mammoth Glacier's advance beyond its modern position at ~1 ka is that Mammoth Glacier doubled in area in a millennium. Mammoth's area today is ~50% of its LIA area—thus at ~1 ka the glacier was of similar size and by 0.1 ka (at the end of the LIA) it reached its maximum Holocene size, i.e.'100%' area. This finding agrees well with the evidence from distal lake sediment cores that Mammoth Glacier was much smaller than its LIA extent from 4.5–1 ka and significantly advanced after 1 ka (Davies, 2011).

Sample MG-04 has exposure ages an order of magnitude lower (10Be: 0.5 ± 0.05 ka, 14C: 0.6 ± 0.01 ka) than the rest of the samples at Mammoth Glacier. The Monte Carlo forward model erosion rate histogram (Figure S6) shows that much higher erosion rates are required for MG-04 than the other samples. Subglacial quarrying is a plausible mechanism for much deeper erosion at one point than others. We interpret that this sample was deeply eroded, probably by quarrying, while the remaining samples are abrasion-dominated. We therefore present two erosion rates from Mammoth Glacier. With MG-04 included, the mean erosion rate is 0.7 ± 0.7 mm yr$^{-1}$. With MG-04 excluded, the mean erosion rate is 0.2 ± 0.2 mm yr$^{-1}$; we consider this more likely to represent an abrasion rate rather than a general erosion (quarrying-included) rate. We plot histograms for both distributions in Figure S7."

- Ln 394 ice dynamics plays a role. . . doesn't ice dynamics always play a role?

    - **Paragraph rewritten, see above.**

- Ln 398 I think this sentence needs to be more precise.

    - **Paragraph rewritten, see above.**

- Ln 414 What about subglacial sediment deposition and insulation from erosion? (Beaud et al 2018, Delaney and Anderson 2022, Stevens et al. 2022).

    - Yes, this is an important consideration. We believe it is addressed on Lines 415–418 of the original manuscript.

- Ln 419 buried. By ice or by debris?

    - Original text on Ln 419 reads "...Conness Glacier buried our sample sites for the duration of the Holocene." Burial by Conness Glacier is meant to imply burial by ice.

- Ln 440–450 I think there is some valuable information in this paragraph. However, much of it is quite general and difficult to link to the paper's findings. Some time distilling this is needed.

    - **We moderate our language with this idea throughout the paper in line with similar comments from Reviewer #2. We change Line 448 to**: "Given this evidence, **we consider the** idea that glaciers advanced past their modern sizes thousands of years apart **because they have experienced non-uniform amounts of modern retreat relative to their LIA extent."**

- Ln 449 their modern positions are more variable that in the past. I am not sure how this conclusion was arrived at.

    - **This has been removed, see above**

- Ln 459 Please describe a quicker glacier.

    - **Changed Line 459 of original manuscript to "...the more quickly a glacier can respond..."**

- Paragraph 462–475 What is the input data for the glacier response time?

    - **We have moved the glacier response time calculations into Methods/Results**. See above reply to general comments. The input data has been moved from a supplemental table to new Table 2, Glacier Hypsometry Data.

- Ln 493 that modern retreat is a reversal of a long-term trend of glacier advance in western North America. . . where do the data presented in this paper show that the glaciers have

advanced, as opposed to simply covered these bedrock areas? is "modern" considered since the LIA?

- **We change the sentence starting on Line 493 to: "Our results provide spatial constraints to understand modern glacier position within the context of the Holocene."**

**Reviewer #2**

This paper details cosmogenic [10]Be and [14]C measurements in bedrock sampled right at the margin of four glaciers across the western US and Canada. The authors apply a Monte Carlo forward modeling technique to determine the most likely timing of glacial advance beyond the modern extent during the Holocene. At face value, the results yield different Holocene glacial histories among the four glaciers, which the authors suggest is unexpected given the uniformity in Holocene climate forcing across the region. To explain this unexpected heterogeneity, the authors hypothesize that the bedrock sampled at each glacier contains information about a unique part of the Holocene ice-margin history because these glaciers are now responding unequally to modern climate change. They attribute the difference in the magnitude of response among these glaciers in part to differences in glacier response time.

I enjoyed learning about the methods employed to interpret the cosmogenic [10]Be and [14]C data, found the figures compelling, and think the proposed hypothesis is interesting. I do have several major questions/comments/concerns that I believe need to be addressed before this work can be published, which I've detailed below. Many of my comments can likely be resolved by tightening up the organization and language. I also think the manuscript would benefit from including more details about some of the methodology and including a more comprehensive literature review in some places. I think these issues can certainly be addressed and hope that the feedback I've provided here is helpful.

Thank you for the opportunity to review this paper and I look forward to seeing where it goes!

*General Comments*

**Methods:** I like the approach used here taking forward modeling with Monte Carlo to estimate the most probable exposure histories represented by the cosmogenic-nuclide concentrations at each of the four glaciers. However, I was left with several questions about the methods used at several points in the paper, in particular in the description of the forward modeling. A lot of the information I was looking for is actually in the paper, but I felt it was buried in table/figure captions or was obscured by confusing (at least to me) language.

**We pull information out of the captions and into the main text wherever possible throughout the manuscript. An example of this change is bringing the Table 2 caption into the main text of Methods 3.1, which now reads:** "All samples were collected in the last five years: JIF Glacier samples collected in 2019; Kokanee Glacier in 2018; Mammoth Glacier in 2020; Conness Glacier in 2018. We present exposure ages in Table 2 (and Figure 5) calculated using the CRONUS-Earth online calculator v.3 (Balco et al., 2008) with the LSDn scaling scheme (Lifton et al., 2014) **and primary production dataset of Borchers et al. (2016)**. These ages assume continuous exposure with no erosion, modern elevation, and standard atmosphere. Uncertainties are analytical (i.e., internal) only."

*Response time calculations, near L103:* How are mean glacier thickness maximum mass loss calculated? I think it is mentioned in the footnote of Table 1 that these values come from other studies. It would be helpful to cite their source in the text when you discuss calculating response time. Also, I think the wording above Equation 1 is backwards? Should it say mean glacier thickness divided by maximum mass loss from the terminus?

**Yes it was backwards and has now been fixed. Second, we appreciate the reviewer's suggestion; the supplementary table has been removed and this data has been brought into the glacier hypsometry table (now called Table 2). We move this section into Methods and Results as per a suggestion from Reviewer #1 and the call for re-organization.**

**On Line 104, we add "Mean glacier thickness is taken from Farinotti et al. (2019) and maximum mass loss at the terminus is from Hugonnet et al. (2021). We omit Conness Glacier from these calculations because there was a large discrepancy between the area of the glacier today and the area of the glacier for which mean thickness was calculated by Farinotti et al."**

**For clarity, we change Line 108 to "…and report the data in Table 2."**

*Apparent exposure age calculations:* There is quite a bit of information buried in the footnotes of Table 2 that might be useful to bring into the text in Section 3.1. In particular, I think information about when you conducted fieldwork and the production rate used in exposure age calculations would be useful to have in the text. In the Table 2 caption, you say that you used the "global" production rate- do you mean the primary production dataset of Borchers et al. (2016), which is the default in v3 of the online exposure age calculator you use? If so, this production rate dataset does not include all production rate calibration data globally, so it might be better to refer to it as "the primary production dataset of Borchers et al. (2016)." Someone corrected me on this recently, I thought I'd pass it along!

**Done. Table 2 caption information brought into main text; "primary production dataset of Borchers et al. 2016" added.**

*Monte Carlo forward modeling and interpretation:* I realize that the methods applied here are described in detail in the Supplement to Vickers et al. (2020), but I think this paper could include a bit more summary information so that the approach taken is clearer. I was left with several questions:

**The comments and questions below were used to help rewrite Methods Section 3.3 detailing the Monte Carlo forward model (see reply to Reviewer #1's general comments).** We provide additional responses here but note that these comments were very helpful in guiding us to better communicate the inner workings of the Monte Carlo forward model.

- How are the burial histories constructed? Is it totally random whether a given timestep has exposure or burial? L208-209 say "the scenarios include a range natural variability intended to simulate natural glacier length oscillations", but I don't really understand how that is implemented.
    - This is helpful, we add this into the revised version of Methods 3.3.
- Is each sample allowed to experience a different amount of erosion? It seems like it based on the last sentence in Section 3.2, but I couldn't tell based on how the method was described. This is important if erosion is used to explain scatter in the datasets.
    - Yes. Hopefully clarified in the new Methods 3.3.
- How were production rates calculated with depth? Basically, what attenuation length was used for spallation and how were muon production rates calculated? There's no mention of production pathways at all, but this is important, especially given the interpretation that some of the higher 14C-10Be ratios at JIF Glacier result from deep glacial erosion.
    - Noted and added to Methods 3.3.
- The authors describe how the probability of exposure was calculated in the caption of Figure 6, but I think it warrants mentioning in the text as well.
    - Added.

- A statement on L278 says, "The probability of bedrock exposure at each site exhibits a quick transition from exposure to burial. For simplicity, we interpret burial to begin at the mid-point of this transition and report the time in which glaciers advanced past modern positions as approximate values," but in the Figure 6 caption it says, "We interpret the change in probability of exposure from greater than 50% to less than 50% to indicate roughly when the glacier advanced past its modern size and bedrock samples were buried". Maybe I am missing something, but I think these are different approaches for interpreting the timing of burial (i.e., none of these probabilities have mid-points at 50%). Based on the reported numbers and looking at Figure 6, I think the wording in the figure caption is more accurate?
    - Done. This was a vestige from a prior draft. We will stick with the 50% line as a reasonable estimate, since otherwise it's hard to pick a particular point. **The line from the caption is now in the main text, and the Line on 278 has been removed.**
- L220: "Samples with 14C-10Be ratios above the production ratio for surface exposure are excluded from the Monte Carlo forward model because they are theoretically impossible, though the possibility of high erosion rates exhuming nuclides produced in the subsurface is explored in the discussion" – It is discussed later in the paper that these ratios can be achieved with deep erosion, so in that context they're not theoretically impossible. Are there model runs that achieve these ratios? If so, do these samples need to be excluded?
    - **We change Line 220 of the original manuscript to: "Samples with $^{14}$C-$^{10}$Be ratios above the production ratio for surface exposure are excluded from the Monte Carlo forward model as some of these samples are not physically reproducible, a decision explored in the Discussion."**
- Or are the erosion rates explored not high enough to achieve these ratios? This comment goes back to my question above about how production rates are calculated because if simplifications were made to the muon production calculation maybe there wouldn't be elevated 14C-10Be ratios in your model results?
- Related to the above comment, the paragraph starting on L349 could maybe use a more technical explanation of why 10Be and 14C production change differently with depth?
- Is it permissible for some samples at a site to be buried while others are not? It seems like not, which is fine, but maybe you could include somewhere the distance of the samples to the 2021 terminus to further justify the assumption that all samples at a site should have the same exposure history.

**The above four bullet points talk about similar concerns in Discussion 5.1 regarding JIF Glacier from Line 335–367 in the original manuscript. We have re-written this section in response to Reviewer #2's comments and provide the new version as follows:**

[revised manuscript text omitted]

**Treatment of Subglacial Erosion:** I appreciate the thorough discussion of subglacial erosion as it is crucial to interpreting the exposure age chronologies. First, I do have a few questions related to the Conness Glacier results are interpreted/presented.

L285-286: I wouldn't consider erosion rates of 0.5 mm/yr to be "exceptionally high." Rather, I think this falls pretty squarely into the range (and maybe even slightly lower than average) expected for mountain glaciers at mid-latitudes? See Cook et al. (2020) titled "The empirical basis for modelling glacial erosion rates", which should be cited here and elsewhere when discussing expected/typical erosion rates (and references therein). In the same paragraph, the authors state, "all three glaciers expected to be more erosive than Conness". Why is this? That statement seems to motivate the "cutoff" value of 0.5 mm/yr as the maximum reasonable erosion rate, so I think it deserves more of an explanation.

Fixed. Reviewer #1 raised a similar concern. Lines 283–287 have been rewritten. See reply to Reviewer #1's general comments.

Based on the figure in the supplement, it's clear that there are an abundance of exposure histories that replicate the nuclide concentrations at Conness when erosion rates are allowed to be higher than 0.5 mm/yr (alluded to in the main text as well), which is why the probability of exposure is <50-60% through the Holocene. I'm just wondering if, rather than burying these results in the supplement, could they be added to Figure 6 as an alternative explanation for the Conness nuclide concentrations, which might make this easier to discuss in the main text as well and explain why it's not the preferred interpretation. It could even be a dotted line to make it clear it's not the preferred interpretation and add that reminder in the figure caption. I don't want this to be confused with over-interpretation of very low nuclide concentrations, so maybe it's not the right solution, but something along these lines may provide some help for the reader in understanding the preferred explanation for these low concentrations.

Yes this is a good point—it's a fine line between trying to be quantitative in our approach and over-interpreting the data. We initially ran the Monte Carlo forward model for all four sites for consistency, but it was never intended to be used on a site with such low concentrations as Conness. **We therefore remove Conness Glacier from Figure 6.** We would rather not over-interpret the Conness Glacier samples by plotting them as a Result with 0% probability of exposure, which we agree seems heavy-handed given that high erosion is a theoretical possibility. We instead only plot our interpretation at Conness Glacier in Figure 7 (the paleoclimate stack plot for the Discussion). We find this more justifiable since it's less of a result and more of an interpretation which can be evaluated by readers. **The new Line 287 reads: "Because of these complexities, we do not present modeling results from Conness Glacier in Figure 5 and explore the various ways to interpret the low concentrations in the Discussion."** Please see reply to Reviewer #1's general comments for the full revised paragraph.

L412-414: "Subglacial quarrying is a plausible mechanism for deep erosion, but quarrying is spatially heterogeneous (Woodard et al., 2019); the measured concentrations at Conness Glacier are remarkably

consistent and clustered tightly around the origin of the isochron plot." I agree with this statement. Something that could help the authors rule out this quarrying scenario (as well as the talus scenario below) could be to put numbers on the range of erosional depths needed (not rates, but total depth of erosion during late Holocene burial) to achieve the measured nuclide concentrations at Conness. This might help with visualization of what size blocks would need to have been removed to yield such low nuclide concentrations (likely large enough that this is not plausible). Also, if the numbers end up being relatively uniform across samples, it's probably unlikely that there were six uniformly large blocks quarried and you just happened to sample at those sites. I think it would be pretty easy to calculate from the model results by multiplying the preferred erosion rate by duration of Late Holocene burial and making another histogram like those already shown in the supplement?

We now present mean erosion rates from the overlapping scenarios in the Abstract and Results.

*One more comment about subglacial erosion:* These results give estimates of subglacial erosion rates at several glaciers across Western North America, which is an important finding because subglacial erosion is something that is difficult to measure because it takes place beneath glaciers. If the authors wanted, they could devote a little space in the discussion to highlighting this finding and placing it in the context of the literature (this is sort of done when stating why the erosion rates at Conness shouldn't be >0.5 mm/yr, but I feel like could first be in a more positive light for the other three glaciers, which may bolster the argument for lower erosion rates at Conness).

We like this suggestion and now present mean erosion rates at three of the four glaciers (as noted above).

**Hypothesis about non-uniform burial durations**
The major conclusion of the paper is that Holocene retreat histories were similar across sites (as the authors expected) but the four glaciers have experienced different magnitudes of modern retreat so that the glaciers that have experienced more modern retreat are now revealing bedrock that spent more of the Holocene covered by ice. This is summarized beginning on L440: "We find the most parsimonious explanation of the non-uniform bedrock burial durations to be that glaciers advanced roughly in concert across North America, as predicted, but have retreated non-uniformly in the industrial era." This is taken one step further to say that "modern glaciers are no longer behaving in concert like they did over the Holocene and are now uniquely out of sync". I think this is an interesting hypothesis but there were a few key things missing for me to completely follow the argument. In some places, it feels like there is some extrapolation beyond the dataset that is not necessary.

First, this argument seems to rely on the set up that all four glaciers are expected to have behaved more or less in concert throughout the Holocene because the "first-order climate controls are similar" (~L60) across the region. The uniformity of Holocene climate change across the Western US and Canada is again stated briefly in the first paragraph of Section 5.2, but for the most part the reader is expected to take these statements at face value. Given that these glaciers span 22º latitude, occupy different climate zones/mountain ranges, and are at different elevations, a slightly more detailed lit review could be given to this topic to establish for the reader that these glaciers should have experienced similar Holocene climate and therefore are expected to have behaved in concert.

The interpretation that modern retreat is tapping into different parts of the Holocene exposure history is really interesting. My issue is the definitive wording surrounding this interpretation. Because this explanation requires that each sampling location yields information about a unique part of the Holocene related to how much the modern glacier has retreated, I don't think you can definitively say that these glaciers did (or didn't) behave synchronously during the Holocene, which is stated several times. It's possible I'm missing something here, but I think the manuscript would benefit from toning down some of the more definitive language in Section 5.3.

Reviewer #2 makes the point that in the absence of specific paleoclimate data from each glacier, we cannot rule out heterogeneous Holocene climate change. We therefore moderate our language on the influence of modern retreat on the non-uniform burial durations. **Lines 29–32 at the end of the original abstract have been removed** ("The intensity and rate of modern warming has exacerbated length differences between glaciers that occur due to hypsometry and response time. We hypothesize that highly varying magnitudes of glacier change in North America today is a departure from similar magnitudes of glacier change over the Holocene.") and **replaced with "We hypothesize that glacier hypsometry, response time, and heterogeneous climate change caused the unequal amounts of retreat."**

**Organization and writing style:** I recognize that my role as a reviewer is to comment on the scientific rigor more than the writing, which is of course up to the authors as approved by the editor. I do feel, however, that this paper would benefit from another pass at organization and wording. For example, I felt the Results section jumped around a bit between burial durations/erosion rates as derived from the two-nuclide diagram and the apparent [10]Be exposure ages. Could results be restructured to first introduce apparent exposure ages, since those are the simplest to describe, then introduce where they plot on two-nuclide diagram? Similarly, the discussion could be organized so that the exposure history of each glacier is established before the non-uniformity of the records is discussed? When discussing the exposure history at each glacier, it could help to start with Konakee and Mammoth because they require the simplest explanations, then go into JIF and Conness which require more complexity. I recognize that there's a lot to cover in this paper and that there's not one right way to organize it, but these are things that as a reader might have worked better for me. I also think a lot of my comments above may have been clarified if some of the wording had been more precise, which will come up in some of my minor comments below.

Thank you for these comments. There is a lot to grapple with in this paper and we spent a lot of time trying to make a logical order out of it (e.g. we initially aimed to be consistent and discuss sites North to South). We appreciate the reviewer's suggestions here. In response, we have moved several sections around to try and make the paper flow better. We sum up our changes in the bullet points at the top of this document.

*Specific comments:*

L19 and elsewhere: "rapidly retreating to their sizes today" – not sure what size means here – length, volume, thickness? I assume lateral extent? This phrasing comes up throughout the paper and could be clarified.

**Noted, will use area instead of size. Changed to** "Glaciers in North America advanced over the Holocene, occupying their maximum Holocene **area** in the late 19th century before rapidly retreating to their **modern area**."

L61: "Glacier length is also a robust record of climate" – does this mean summer temperature?

This paragraph has been adjusted to add clarity. We do use 'climate changes' as an intentional catch-all here because that is the language in the reference, and there is debate about the precise influence of temperature, precipitation, and seasonality.

L40: "how" is vague, does this mean the degree to which glaciers fluctuated? Response of glaciers to past warming/cooling?

**Done. Changed this in Ln 40 of original manuscript to "Glacier length fluctuations in the Holocene provide…" as per a similar comment from Reviewer #1.**

L92-94: The authors state that glaciers across region began retreating from LIA moraines 1880 CE, but don't say how that is known (like, what did the authors in the cited study do, what is the evidence, is it really that all glaciers in the western US began retreating by 1880?). Could there be a bit more explanation? In general throughout the paper I feel like there are statements like this (usually with citations, which is great!), but as a reader who is not an expert on when glaciers in the western US retreated from their LIA moraines, I just was looking for a bit more info about the literature. I'd just keep an eye out for these!

Fixed and noted, please see the new version of Section 2 (now called Background) in the reply to Reviewer #1's general comments.

L105: "we caveat this response" – I'm not sure caveat can be used as a verb like this?

**Done. Sentence now reads "We note this calculation is a minimum estimate of how quickly a glacier could adjust its volume because we use the maximum mass loss from the terminus; the purpose of this calculation is merely to serve as a common point of comparison amongst the four glaciers, rather than a robust estimate of glacier response time."**

L111: "lowest-sloping ice surface" is confusing – maybe, surface slope of only 5º, or, shallowest/gentlest slope?

**Done.**

L111: units for response time. Years?

**Done.**

L111-112: "most impacted by absolute area loss from climate change, but least impacted by percentage relative to its Holocene maximum at 70% of its LIA area" – I'm not sure I understand this. Absolute area has changed the most since the end of the LIA but still occupies 70% of its LIA extent? The phrasing, "LIA area loss", used throughout Section 2, is confusing to me. Does this mean the glacier in 2021 was 70% of its LIA footprint?

**Sentence has been changed to "It has retreated the least relative to its LIA area."**

L 112-113: "2003 area of JIF glacier mapped in Figure 1 is likely similar to area of the glacier at ~2 ka prior to the latest Holocene and LIA advances" – Is this stated in Clague? If so, could use more of a (brief) summary of what they did and how they came to this conclusion.

**Done. See revised version of the Background in reply to Reviewer #1's general comments.**

L143: Table 1 says that response time couldn't calculated for Conness because of a lack of suitable data (which I assume to be mean thickness and maximum mass loss?). Paragraph here says "RGI mapping is too different from glacier size today". I don't understand what that means and why that precludes calculating response time. Also, could add something like "given the relationship between surface slope and response time, likely has response time similar to Kokanee Glacier"?

**We change the sentences on Line 143 to:** "Conness has the steepest ice surface slope, 23°, suggesting the quickest response time of the glaciers studied here given the relationship proposed by Zekollari et al. (2020). We omit Conness Glacier from these calculations because there was a large discrepancy between the area of the glacier today and the area of the glacier used in the mean thickness calculations by Farinotti et al."

L182: Several samples, or several samples at each glacier?

**Done. Changed to:** "We collected several (n = 5–9) bedrock samples **at each glacier** that were recently exposed during modern retreat (see supplement for field photos and historical imagery)."

L183: "we assume that all samples [within a catchment, at each glacier] experienced…"?

**Done. Similar correction requested by Reviewer #1, sentence now reads "**We assume that all samples **at a given glacier** experienced…"

L263-264: "the Conness glacier … four glaciers preseted here" – these sentences basically say the same thing. Could condense.

**Done.**

L297: "results at all four glaciers best explained by glacier expansion from the early to late Holocene" – continuous expansion from the early to late Holocene? I don't think these data show whether that glaciers were expanding or retracting through the entire Holocene, but do show that three of the glaciers were generally smaller than today in the early-to-mid Holocene and larger than today in the late Holocene.

We don't mean continuous here, simply that the glaciers expanded from some area in the early Holocene to some size in the late Holocene. Conness has been removed from this interpretation as well. The sentence on Line 297 now reads: "…our results at three of four sites suggest glacier expansion from early to late Holocene." (This is shown in reply to Reviewer #1; copying again here for clarity.)

General comment about presentation of apparent exposure ages: how is the cutoff of "three highest concentration samples" determined? Also, is this truly highest concentrations or oldest ages? Possible it's the same for these sites because elevations/shielding are likely similar?

**Done, added in a note about our logic on Line 233 of the original manuscript:**

"We plotted the apparent exposure ages from each sample onto satellite imagery of the glacier forefields (Figure 4). We note that glacial erosion has removed surface bedrock non-uniformly, and thus the ages are only apparent ages until erosion depths are constrained. The highest ages are likely to have experienced the least erosion and thus provide a rough sense for 'true' Holocene exposure. We therefore calculated the mean and standard deviation of the three highest apparent ages from each glacier to approximate Holocene exposure of the bedrock transect."

L303: Could use more context about the increasing glaciogenic sediment flux in Konrad and Clark, 1998 – how long is the record? Is there glaciogenic sediment the entire time? Could give more space to this comparison, as it seems like these data could potentially help support the preferred interpretation.

**Done. Added to Background.**

L367: What does minimum estimate mean here? That JIF likely advanced earlier than 2 ka?

**This statement has been removed as part of rewrite on Lines 335–367.**

L376: "retreat events did not extend beyond modern position of the glacier" – should this be that the glacier was not smaller than today during those retreat episodes?

Done. This line was rewritten in response to a similar comment from Reviewer #1. **The sentence now reads: "Our results suggest the glacier was expanded beyond its position today from ~6 ka through the LIA, and that any glacier length fluctuations happened between the position occupied at ~6 ka and its LIA maximum extent.**

L389: What does it mean for "the low 14 C10 Be ratio of MG-02 to be an artifact of the simplifications made for the isochron plot"?

**We have removed this sentence; see rewrite of Lines 378–398 in reply to Reviewer #1.**

L454: I thought the preferred interpretation was that Conness glacier samples were buried the whole Holocene. How did it recede past its minimum size of the Holocene? Wouldn't that imply exposure at some point?

The samples were buried until modern-day exposure (and sampling).

L462-463: "Kokanee Glacier has retreated 46% from its LIA extent, an intermediate amount between Conness (14%) and JIF (70%) Glaciers." This makes it seem like Conness retreated the least and JIF retreated the most compared to its LIA extent? I think it's the opposite?

**Done. Changed to: "Kokanee Glacier is 46% of its LIA extent,** an intermediate amount between Conness (14%) and JIF (70%) Glaciers."

L458: What does "nearly the least relative to its Holocene advance history" mean?

**Done. Changed to "receded only to a size last occupied at ~2 ka."**

L459-461: This is an important sentence but I'm not sure I understood it – what is "the further back into its Holocene advance history it resides today" – revealing rock that spent less of the Holocene covered?

**These two comments address a similar point. We have clarified the paragraph on Lines 451–461 and it now reads:**

"In support of this hypothesis, there appears to be a connection between the amount of retreat relative to LIA extent, glacier response time, and the exposure-burial durations. Conness likely has the fastest response time of the four glaciers (smallest area; steepest slope) and thus we expect it would retreat the most relative to its LIA extent and reveal bedrock that has been buried the longest, which is what we observe; Conness is the most retreated relative to its LIA extent and the only glacier to expose bedrock that has been buried for the duration of the Holocene (burial duration > ~11 ka). JIF Glacier, on the other hand, is two orders of magnitude larger, has a lower slope, and has the slowest approximate response time we calculated ($\tau$ = 27 years); we expect it would retreat the least relative to its LIA extent and reveal bedrock that has only been recently buried, which again is largely what we observe where it has retreated the least relative to its LIA extent and has receded only to a size last occupied at ~2 ka. Taken together, a simple relationship emerges: the quicker a glacier can respond to industrial-era climate change, the more

it has retreated relative to its LIA extent, the longer that bedrock has been buried as it is being revealed by modern retreat."

This edit also addresses revisions requested by Reviewer #1 on the same paragraph.

Table 1: Does it make sense to provide mean glacier thickness and mass loss at terminus? I realize that these are from other studies and are included in a supplement al table, but that supplementary table could probably be eliminated by including these values in Table 1.

**Done. Moved into Table 1.**

**Technical corrections:**

L23 and elsewhere: minor, but I believe Icefield is all one word?

**Done.**

L90: moraines plural?

**Done.**

L191: I think spatially should just be spatial?

**Done.**

L370: inferring should be inferred?

**Done.**

---

## Referee Report (RR1)

**Review of "Four North American glaciers advanced past their modern positions thousands of years apart in the Holocene"**

October 16, 2023

**Letter**

Dear Editor,

Upon re-reading the manuscript, it is clear that the authors have put substantial consideration into the comments from both reviewers, and the manuscript is very much improved.

From my perspective, there are minimal scientific comments regarding the manuscript at this point. However, in my opinion there are many places where the language can be imprecise and the writing can be clarified or condensed. Again, these are matters of discretion for the editor and authors, however, I believe that by addressing these issues the quality of the paper will increase substantially. The list below includes some places where improvement is likely needed. But, it is not exhaustive and additional changes might be needed. There are also a few minor scientific comments and suggestions.

With these comments, minor revisions are likely necessary, and I do not believe I need to see the manuscript again. However, I am available as the editor wishes.

Again, hopefully these comment prove useful and congratulate the authors on there near-completion of this work.

**Specific comments**

- Line 59: "We attempt to minimize the complexities of comparing glaciers to one another by focusing on glacier area change rather than area alone, which should control for the differences in hypsometry and background climate.": I am not sure what is meant precisely by this comment as both area and area change are a result of hyposmetry and background climate.

- Line 67: "more heterogeneity than expect": what would you expect? some rephrasing needed.

- Line 128: What does "internal" mean?

- Figure 2: Would this figure be improved if a title was added to each row for instance *Exposure* for the first, *Burial* for the second, and *Re-exposure* for the third? That way readers can easily translate the process in the figure to the terminology in the text.

- Paragraph at Line 174: Because the Monte Carlo method is discussed, I would recommend making the two parameters evaluated very explicit here.

- Line 177: I am not sure how this list can be "exhaustive" especially given the parameters shown in the paragraph starting at 199 and the temporal discretisation presented in the model. Is it necessary to include? Also, if it is an exhaustive list, would this be a "grid search"? If so, explaining this would improve the model.

- Line 178: "the model calculates how 14C and 10Be concentrations evolve" change to *the model calculates the evolution of 14C and 10Be concentrations*. Similar issues exist in other parts of the text.

- Paragraph at Line 196: Much of this content was presented in the first paragraph of the section and is closely linked with the next paragraph. Some reorganization of the section might be needed.

- Line 213: "Scenarios that reproduce all nuclide concentrations for all samples at a given glacier are recorded and saved as viable exposure-burial histories" why not write *Scenarios reproducing all nuclide concentrations for all samples at a given glacier deemed as viable exposure-burial histories*.

- Line 217: "overlapping" this term is repeatedly used through out the text. However, I have not found a clear definition of it. To me it seems visually evident in plots in the supplement. However, this needs to be clearly described here, and forgetful readers (like myself), might need a little reminder of what it means when it is discussed in the results.

- Line 230: "The RGI data are from miscellaneous years based on what was available in RGI." This sentence must be more precise or excluded.

- Line 249: "Apparent exposure ages are presented in Table 1 and plotted onto satellite imagery of the glacier forefields in Figure 3." I noticed that the authors seem to include sentences like this at the beginning of paragraphs and sections describing a Figure or Table. The text will be far shorter and more interesting if these comments are omitted. It should be enough just to reference the figure in the text.

- Paragraph at Line 249: it is not clear to me if these are ages from 10Be or 14C. or both. Please describe.

- Line 257: see comment for Line 249.

- Table 1: Would it be helpful to make a row for each glacier that has its average of each column for that glacier? This may make it much easier for readers to extract the key points or messages from the table.

- Line 273: "The second observation is that 14C-10Be sample ratios at JIF Glacier, Kokanee Glacier, and Mammoth Glacier are depressed." Depressed compared to what?

- Line 278-279: "Low ratios are only possible through extensive decay of 14C relative to 10Be, severely limiting the number of plausible scenarios." maybe change plausible scenarios to *plausible climate/glacier scenarios*. Otherwise, a nice sentences that concisely describes an trend in the data. Consider moving to the top of the paragraph, so that it is in the readers' minds as they interpret you findings.

- Section 4.2: Some comments about the meaning of the results might help in this section.

- Line 295: "We plotted. . . ". See comment for Line 249.

- Line 308 - 310: Super interesting. I believe Reviewer 2 made some comments about this in the last draft and to me it is great that the authors pursued this further. One potential thought that I had was to compare these results to "modern erosion rate" from Cook et al., 2020 (which the authors do), then discuss the findings in terms of finding by Ganti et al., 2016 (Time scale bias in erosion rates of glaciated landscapes), for instance. I hope this is not too much of a tangent (more an idea). If it is, please ignore.

- Figure 5: I would recommend adding "Ice Free" to the axis underneath the "Probability of Exposure." This may reduce some confusion for more "glaciology" focused readers may not thinking in terms of "exposure."

- Line 340: "It has also retreated more than the composite of Sierra Nevada glaciers." this needs a citation.

- Table 2: Can this table be organized so that the two most important quantities are right next to each other? For me, the two most important bits of information are response time and %LIA Area. It will make it easier to compare these values.

- Line 374: Why would this be surprising?

- Line 396–398: This is a very complicated sentence. Simplify?

- Line 427: "then it should be replicable by modeling". Can the section or figures that show how this was not replicated by modeling be shown or referenced?

- Figure 8 caption: "Erosion rates are marked by red vertical tick marks along each scenario (black dashed line), starting from an erosion rate of 0.0 mm yr-1 on the righthand side progressing to 5.0 mm yr-1." Very nice way to plot this. However, it took me a little bit to understand. Can this be rephrased? would it help to remove some isochrons? No problem if not.

- Lines 477-482: Dirk Sherler, amongst others, has some papers about this. Comments would be improved if this was supported by some observations and citations.

- Line 484: "This interpretation is plotted in Figure 7. . . " This sentence is a bit confusing and can be omitted by just referencing Figure 7.

- Paragraph at 529: This is probably correct, strictly speaking, but a little bit hard to follow and understand the main message. If the exact figures are needed, then the Table can simply be referenced.

---

## Author Response (AR2)

**2nd Round Revisions.**

We thank Reviewer #1 for their time in reviewing our manuscript again. Our comments are in blue.

**Reviewer #1 (sole reviewer)**

Dear Editor,

Upon re-reading the manuscript, it is clear that the authors have put substantial consideration into the comments from both reviewers, and the manuscript is very much improved.

From my perspective, there are minimal scientific comments regarding the manuscript at this point. However, in my opinion there are many places where the language can be imprecise and the writing can be clarified or condensed. Again, these are matters of discretion for the editor and authors, however, I believe that by addressing these issues the quality of the paper will increase substantially. The list below includes some places where improvement is likely needed. But, it is not exhaustive and additional changes might be needed. There are also a few minor scientific comments and suggestions.

With these comments, minor revisions are likely necessary, and I do not believe I need to see the manuscript again. However, I am available as the editor wishes.

Again, hopefully these comment prove useful and congratulate the authors on there near-completion of this work.

**Specific comments**

- Line 59: "We attempt to minimize the complexities of comparing glaciers to one another by focusing on glacier area change rather than area alone, which should control for the differences in hypsometry and background climate.": I am not sure what is meant precisely by this comment as both area and area change are a result of hyposmetry and background climate.
    - Changed to "We attempt to minimize the complexities of comparing glaciers to one another by focusing on glacier area change, which should control for the hypsometric variables that remain roughly constant over the Holocene (e.g. elevation, bed-slope, aspect) and background climate conditions (high-versus-low precipitation regime)."

- Line 67: "more heterogeneity than expect": what would you expect? some rephrasing needed.
    - We provide further clarity now and our sentence reads: "Our findings indicate more heterogeneity in glacier length amongst the four sites than expected given their relative proximity to one another and shared climate forcings."
    - The argument here is that they're relatively close together and experienced similar forcings, so we expect a similar response. On line 54, we write "It is expected that glaciers across western North America advanced roughly synchronously over the Holocene because the first-order climate controls are similar:"

- Line 128: What does "internal" mean?
    - Changed to "Uncertainties presented are analytical (i.e., internal; measurement) only." Here we provided this additional clarity because CRONUS ages as presented by the online calculator output have an 'internal' uncertainty and 'external' uncertainty reported.

- Figure 2: Would this figure be improved if a title was added to each row for instance Exposure for the first, Burial for the second, and Re-exposure for the third? That way readers can easily translate the process in the figure to the terminology in the text.
  - Done.

- Paragraph at Line 174: Because the Monte Carlo method is discussed, I would recommend making the two parameters evaluated very explicit here.
  - Based on the prior suggestions by both reviewers, we feel that we have made it clear in this section what parameters are being modeled with our Monte Carlo analysis.

- Line 177: I am not sure how this list can be "exhaustive" especially given the parameters shown in the paragraph starting at 199 and the temporal discretisation presented in the model. Is it necessary to include? Also, if it is an exhaustive list, would this be a "grid search"? If so, explaining this would improve the model.
  - We have deleted 'exhaustive' to avoid confusion.

- Line 178: "the model calculates how 14C and 10Be concentrations evolve" change to the model calculates the evolution of 14C and 10Be concentrations. Similar issues exist in other parts of the text.
  - Done.

- Paragraph at Line 196: Much of this content was presented in the first paragraph of the section and is closely linked with the next paragraph. Some reorganization of the section might be needed.
  - Based on the prior suggestions by both reviewers, we extensively redrafted this section from the original draft and feel this section is clear as constructed. The redundancy is recognized but we think is necessary for continued clarity based on prior comments.

- Line 213: "Scenarios that reproduce all nuclide concentrations for all samples at a given glacier are recorded and saved as viable exposure-burial histories" why not write Scenarios reproducing all nuclide concentrations for all samples at a given glacier deemed as viable exposure-burial histories.
  AND
- Line 217: "overlapping" this term is repeatedly used through out the text. However, I have not found a clear definition of it. To me it seems visually evident in plots in the supplement. However, this needs to be clearly described here, and forgetful readers (like myself), might need a little reminder of what it means when it is discussed in the results.
  - We agree with both comments and now on Line 211 we add "Scenarios that reproduce all nuclide concentrations for *all* samples at a given glacier are recorded and saved as viable exposure-burial histories; we henceforth refer to these as the 'overlapping' scenarios."

- Line 230: "The RGI data are from miscellaneous years based on what was available in RGI." This sentence must be more precise or excluded.
  - We agree and have deleted this sentence. It is clear from prior sentence where we say "We also included an intermediate position using the outlines available from the Randolph Glacier Inventory."

- Line 249: "Apparent exposure ages are presented in Table 1 and plotted onto satellite imagery of the glacier forefields in Figure 3." I noticed that the authors seem to include sentences like this at the beginning of paragraphs and sections describing a Figure or Table. The text will be far shorter

and more interesting if these comments are omitted. It should be enough just to reference the figure in the text.

- • Again, given prior reviewer feedback that clarity was needed throughout, we have elected to leave this section as-is since these additions were made from prior reviewer comments.

- Paragraph at Line 249: it is not clear to me if these are ages from 10Be or 14C. or both. Please describe.
  - • We agree and have added '$^{10}$Be' throughout the paragraph now.

- Line 257: see comment for Line 249.
  - • see our comments above

- Table 1: Would it be helpful to make a row for each glacier that has its average of each column for that glacier? This may make it much easier for readers to extract the key points or messages from the table.
  - • We present the means in the text and prefer to leave this as is for clarity.

- Line 273: "The second observation is that 14C-10Be sample ratios at JIF Glacier, Kokanee Glacier, and Mammoth Glacier are depressed." Depressed compared to what?
  - • Done. Added: '…relative to the production ratio.'

- Line 278-279: "Low ratios are only possible through extensive decay of 14C relative to 10Be, severely limiting the number of plausible scenarios." maybe change plausible scenarios to plausible climate/glacier scenarios. Otherwise, a nice sentences that concisely describes an trend in the data. Consider moving to the top of the paragraph, so that it is in the readers' minds as they interpret you findings.
  - • Done. Changed to: "… severely constraining the plausible exposure-burial scenarios."

- Section 4.2: Some comments about the meaning of the results might help in this section.
  - • Since the model is statistical and only outputs exposure and burial scenarios, we prefer to leave any interpretation to the Discussion for consistency and clarity.

- Line 295: "We plotted. . . ". See comment for Line 249.
  - • see our comments above

- Line 308 - 310: Super interesting. I believe Reviewer 2 made some comments about this in the last draft and to me it is great that the authors pursued this further. One potential thought that I had was to compare these results to "modern erosion rate" from Cook et al., 2020 (which the authors do), then discuss the findings in terms of finding by Ganti et al., 2016 (Time scale bias in erosion rates of glaciated landscapes), for instance. I hope this is not too much of a tangent (more an idea). If it is, please ignore.
  - • Thank you. Yes agreed it's super interesting. For fear of getting tangential in the manuscript, we leave it as is, but we do mention that the erosion rates we find are on the low end of published estimates (see discussion about JIF Glacier and the conclusion).

- Figure 5: I would recommend adding "Ice Free" to the axis underneath the "Probability of Exposure." This may reduce some confusion for more "glaciology" focused readers may not thinking in terms of "exposure."
  - • Since 'ice-free' is an interpretation (versus simply 'exposed') we prefer to leave it as is. Again, we understand the reviewers point but we are trying to be clear and consistent about what the model is really outputting versus our interpretation.

- Line 340: "It has also retreated more than the composite of Sierra Nevada glaciers." this needs a citation.
  - Done. Now we refer to Figure 6 now where it is plotted and cited in the caption.

- Table 2: Can this table be organized so that the two most important quantities are right next to each other? For me, the two most important bits of information are response time and %LIA Area. It will make it easier to compare these values.
  - It is currently laid out with respect to how the calculations are made, and it is not clear that putting them next to each other adds clarity. We have elected to leave the table as-is simply for ease and clarity.

- Line 374: Why would this be surprising?
  - As outlined in introduction and preceding paragraph, we expected similarity across sites.

- Line 396–398: This is a very complicated sentence. Simplify?
  - Done. Re-written as: "An interesting implication therein is that Mammoth Glacier doubled in area in less than a millennium: at ~1 ka the glacier's area was 1.7 $km^2$ and at ~0.1 ka (end of the LIA) it was 3.4 $km^2$."

- Line 427: "then it should be replicable by modeling". Can the section or figures that show how this was not replicated by modeling be shown or referenced?
  - This sentence just leads into the next paragraph where we do some modeling and explain this in more detail.

- Figure 8 caption: "Erosion rates are marked by red vertical tick marks along each scenario (black dashed line), starting from an erosion rate of 0.0 mm yr-1 on the righthand side progressing to 5.0 mm yr-1." Very nice way to plot this. However, it took me a little bit to understand. Can this be rephrased? would it help to remove some isochrons? No problem if not.
  - This certainly is a complicated plot. We deliberated for a while on how to best present it and are very happy that you found it helpful. Given how tricky it was to convey the information, we elect to leave the figure as we have it since based on experience presenting our work this appears to be the clearest figure for other researchers to understand.

- Lines 477-482: Dirk Sherler, amongst others, has some papers about this. Comments would be improved if this was supported by some observations and citations.
  - These papers seem to be about debris cover on ice impacting a glacier's response to climate change, rather than about how debris cover may have impacted nuclide concentrations in a glacier's forefield. We agree a citation would be helpful; we have not found anything else in the literature that deals with our specific problem of debris cover impacting cosmogenic nuclide concentrations in a glacier forefield.

- Line 484: "This interpretation is plotted in Figure 7. . . " This sentence is a bit confusing and can be omitted by just referencing Figure 7.
  - Changed to: Without erosion rates > 0.5 mm $yr^{-1}$, there are no overlapping scenarios with exposure from our Monte Carlo forward model, and the modeled probability of exposure is '0%' (Figure 7).

- Paragraph at 529: This is probably correct, strictly speaking, but a little bit hard to follow and understand the main message. If the exact figures are needed, then the Table can simply be referenced.
    - Done. The numbers were removed for clarity and the Table was referenced.